# Fetal liver macrophages contribute to the hematopoietic stem cell niche by controlling granulopoiesis

Amir Hossein Kayvanjoo[1], Iva Splichalova[1], David Alejandro Bejarano[2], Hao Huang[1], Katharina Mauel[1], Nikola Makdissi[1], David Heider[1], Hui Ming Tew[1], Nora Reka Balzer[1], Eric Greto[3,4], Collins Osei-Sarpong[5], Kevin Baßler[6], Joachim L Schultze[6,7,8], Stefan Uderhardt[3,4], Eva Kiermaier[9], Marc Beyer[5,7,8], Andreas Schlitzer[2], Elvira Mass[1]*

[1]Developmental Biology of the Immune System, Life & Medical Sciences (LIMES) Institute, University of Bonn, Bonn, Germany; [2]Quantitative Systems Biology, Life & Medical Sciences (LIMES) Institute, University of Bonn, Bonn, Germany; [3]Department of Internal Medicine 3-Rheumatology and Immunology, Deutsches Zentrum für Immuntherapie (DZI) and FAU Profile Center Immunomedicine (FAU I-MED), Friedrich Alexander University Erlangen-Nuremberg and Universitätsklinikum Erlangen, Erlangen, Germany; [4]Exploratory Research Unit, Optical Imaging Centre Erlangen, Erlangen, Germany; [5]Immunogenomics & Neurodegeneration, Deutsches Zentrum für Neurodegenerative Erkrankungen (DZNE), Bonn, Germany; [6]Genomics & Immunoregulation, LIMES Institute, University of Bonn, Bonn, Germany; [7]Systems Medicine, Deutsches Zentrum für Neurodegenerative Erkrankungen (DZNE), Bonn, Germany; [8]PRECISE Platform for Single Cell Genomics and Epigenomics, DZNE and University of Bonn, Bonn, Germany; [9]Immune and Tumor Biology, Life & Medical Sciences (LIMES) Institute, University of Bonn, Bonn, Germany

*For correspondence:
emass@uni-bonn.de

Competing interest: The authors declare that no competing interests exist.

**Abstract** During embryogenesis, the fetal liver becomes the main hematopoietic organ, where stem and progenitor cells as well as immature and mature immune cells form an intricate cellular network. Hematopoietic stem cells (HSCs) reside in a specialized niche, which is essential for their proliferation and differentiation. However, the cellular and molecular determinants contributing to this fetal HSC niche remain largely unknown. Macrophages are the first differentiated hematopoietic cells found in the developing liver, where they are important for fetal erythropoiesis by promoting erythrocyte maturation and phagocytosing expelled nuclei. Yet, whether macrophages play a role in fetal hematopoiesis beyond serving as a niche for maturing erythroblasts remains elusive. Here, we investigate the heterogeneity of macrophage populations in the murine fetal liver to define their specific roles during hematopoiesis. Using a single-cell omics approach combined with spatial proteomics and genetic fate-mapping models, we found that fetal liver macrophages cluster into distinct yolk sac-derived subpopulations and that long-term HSCs are interacting preferentially with one of the macrophage subpopulations. Fetal livers lacking macrophages show a delay in erythro-poiesis and have an increased number of granulocytes, which can be attributed to transcriptional reprogramming and altered differentiation potential of long-term HSCs. Together, our data provide a detailed map of fetal liver macrophage subpopulations and implicate macrophages as part of the fetal HSC niche.

## Editor's evaluation

Using single-cell sequencing, high-resolution imaging, and inducible genetic deletion of yolk-sac (YS) derived macrophages, the authors present a useful map of fetal liver macrophage subpopulations and provide important data demonstrating that heterogeneous fetal liver macrophages regulate erythrocyte enucleation, interact physically with fetal HSCs, and may regulate neutrophil accumulation in the fetal liver. These important findings provide a solid foundation for further investigating the effects of macrophages on HSC function during fetal hematopoiesis and into adulthood and will be useful for the field of macrophage biology and developmental hematopoiesis.

## Introduction

Macrophages are found in all adult organs, where they perform essential functions during inflammatory responses as well as in tissue homeostasis, such as tissue remodeling, phagocytosis of apoptotic cells, and production of cytokines and growth factors. Work in mice showed that most macrophages originate from erythro-myeloid progenitors (EMPs) in the yolk sac and are long lived, and that their maintenance in many adult tissues does not rely on definitive hematopoiesis (*Gomez Perdiguero et al., 2015*; *Hoeffel et al., 2015*; *Mass, 2018*). All developing tissues are initially colonized by circulating pre-macrophages (pMacs), which immediately differentiate into tissue-specific macrophages (*Mass et al., 2016*; *Stremmel et al., 2018*), therefore, macrophages are an integral part of organogenesis.

In adult tissues, macrophages inhabit distinct anatomical niches within organs, for example, the lung harbors alveolar and interstitial macrophage populations (*Aegerter et al., 2022*), while macrophages in the adult liver are divided into Kupffer cells, liver capsular, central vein, and lipid-associated macrophages (*Guilliams and Scott, 2022*). Within these niches, resident macrophages adapt to their tissue environment and perform specific tasks to maintain organ function, such as mucus clearance by alveolar macrophages or phagocytosis of red blood cells by Kupffer cells (*Mass et al., 2023*). However, the role of macrophages in organ development and function as well as their heterogeneity during embryogenesis is less well understood.

One of the few well-known functions of fetal macrophages is their involvement in erythroblast maturation. In the fetal liver, erythroblastic island (EI) macrophages serve as niches for erythroblasts. During erythropoiesis, primitive and definitive erythroblasts directly interact with EI macrophages, where they undergo final steps of maturation, including expelling of their nucleus, which is phagocytosed by EI macrophages (*Palis, 2016*; *Palis, 2014*). Previous studies identified EI macrophage heterogeneity in the fetal liver at embryonic day (E)13.5/E14.5 (*Li et al., 2019*; *Mukherjee et al., 2021*; *Seu et al., 2017*). EI macrophages were shown to express different levels of cell adhesion proteins such as Vcam1, CD169, and CD163, as well as other proteins that are important for their function as EI niche (e.g., Epor, Klf1, EMP, and DNaseII), thereby promoting erythropoiesis (*Li et al., 2019*; *Mariani et al., 2019*; *May and Forrester, 2020*; *Mukherjee et al., 2021*). In addition to erythropoiesis, the E13.5/E14.5 fetal liver is at the peak of hematopoiesis, providing a niche for the expansion and differentiation of other hematopoietic stem and progenitor cells (HSPCs) (*Lewis et al., 2021*). Thus, the liver is a complex cellular interaction network where the role of macrophages in other hematopoietic developmental processes, such as myelopoiesis, is not fully understood.

Recent evidence suggests that one of the core macrophage functions is the support of stem and progenitor cell functionality in different tissues. For instance, pericryptal macrophages in the gut interact with epithelial progenitors, thereby promoting their proliferation and differentiation via Wnt, gp130, TLR4, or NOX1 signaling (*Delfini et al., 2022*). In the bone marrow, distinct macrophage subpopulations contribute directly and indirectly to the hematopoietic stem cell (HSC) niche. Osteoclasts are a highly specialized EMP-derived macrophage population responsible for bone resorption, whose function is necessary to generate the bone marrow and, thereby, the postnatal HSC niche (*Jacome-Galarza et al., 2019*). Co-culture experiments using different combinations of hematopoietic cells indicate that osteomacs support the HSC niche in synergy with megakaryocytes (*Mohamad et al., 2017*). DARC[+] macrophages are in direct contact with CD82[+] long-term (LT)-HSCs in the endosteal and arteriolar niches, where they contribute to LT-HSC dormancy via maintenance of CD82 expression (*Hur et al., 2016*). Additional macrophage depletion studies via clodronate and diphtheria toxin indicate that CD169[+] macrophages in the bone marrow promote HSC retention by acting specifically on the Nestin[+] HSC niche (*Chow et al., 2011*), as well as steady-state and stress erythropoiesis (*Chow et al., 2013*).

During embryogenesis, macrophages have been shown to be important for the development and proliferation of hematopoietic stem- and progenitor cells in mice. In the mouse, the first HSCs develop from the aorta–gonad–mesonephros (AGM) region starting at E10.5. Here, a CD206+ macrophage population actively interacts with nascent and emerging intra-aortic HSCs (*Mariani et al., 2019*). Macrophage depletion studies via clodronate liposomes and the Csf1r inhibitor BLZ945 using AGM explants indicate that the presence of macrophages in the AGM is influencing HSC production (*Mariani et al., 2019*). Furthermore, co-culture studies of a mixture of hematopoietic stem- and multipotent-progenitor cells (HSC/MPP) with macrophages isolated from the fetal liver as well as clodronate depletion studies during embryogenesis suggest that fetal liver macrophages promote HSC/MPP proliferation (*Gao et al., 2022*). Indeed, immunofluorescent stainings of F4/80+ macrophages and CD150+ cells (*Gao et al., 2022*) indicate that macrophages could also be part of the LT-HSC niche in the fetal liver, thereby influencing the proliferation and differentiation of stem cells at the top of the HSC hierarchy.

Studying stem cell niches and defining the molecular factors that promote stem cell proliferation and differentiation are essential to build a robust and reproducible in vitro system that may serve as a universal source of functional stem cells and/or immune cells for therapeutic purposes. This has already been achieved for induced pluripotent stem cells (iPSCs), which can be infinitely expanded and differentiated into the cell type of interest, thereby representing a safe product to treat diseases (*Yamanaka, 2020*). In contrast, a universal and well-defined culturing protocol for LT-HSCs allowing for continuous expansion is lacking (*Kumar and Geiger, 2017*; *Wilkinson et al., 2020*). Production of LT-HSCs from iPSCs represents a promising alternative, however, differentiation protocols without ectopic transcription factor expression have not been established yet (*Demirci et al., 2020*). This may be due to the complex developmental programming of adult LT-HSCs that have experienced distinct niche signals during their migration from AGM via the fetal liver to the bone marrow, resulting in functional differences observed in fetal versus mature HSCs (*Arora et al., 2014*). Growing evidence, mainly provided by in vitro and ex vivo studies in combination with macrophage depletion via clodronate (*Gao et al., 2022*; *Mariani et al., 2019*), suggests that macrophages play an essential role in HSC development and maintenance during embryogenesis. However, it remains to be investigated whether specific macrophage populations in the fetal liver contribute to LT-HSC stem-cell ness or differentiation via factors they produce in vivo.

Given the observations implicating macrophages in HSC functionality, we characterized the heterogeneity and ontogeny of fetal liver macrophage populations, providing a comprehensive macrophage atlas of the fetal liver at E14.5. Furthermore, using a conditional mouse model to deplete macrophages in vivo, we established fetal liver macrophages as important modulators of LT-HSC differentiation capacity into granulocytes.

## Results

### The fetal liver harbors heterogeneous macrophage populations

To investigate fetal liver macrophage heterogeneity, we performed single-cell RNA-sequencing on sorted CD11b$^{low/+}$ cells at E14.5 (*Figure 1—figure supplement 1A*). To isolate macrophages and macrophage progenitors for further downstream analyses from this myeloid population, we clustered all cells and overlaid a pMac signature (*Mass et al., 2016*). Out of eleven clusters, four clusters were chosen (*Figure 1—figure supplement 1B, C* , see Methods) and analyzed further. To further select specifically macrophages, the same procedure was performed twice on the re-clustered cells using a signature enriched in fetal macrophages when compared with EMPs (*Figure 1—figure supplement 1D*), resulting in 18 clusters (*Figure 1—figure supplement 1E*). Cells that either expressed macrophage precursor genes (*Clec7a*, *Ccr2*, *Cx3cr1*, *Csf1r*), pan-macrophage genes (*Mrc1*, *Adrgre1*, *Siglec1*, *Msr1*, *Cd63*), and/or liver macrophage-specific genes (*Timd4*, *Clec4f*, *Vcam1*) were enriched in clusters 1, 2, 7, 8, 9, and 11, which were chosen for further downstream analysis (*Figure 1A, B*, *Figure 1—figure supplement 1F*). The predominant expression of *Ly6c2*, *Ly6g*, *Cxcr2*, and *Cd33* in the remaining clusters (*Figure 1—figure supplement 1F*) indicated their monocyte/granulocyte or myelomonocytic precursor cell identity, respectively, and were therefore excluded from further analysis.

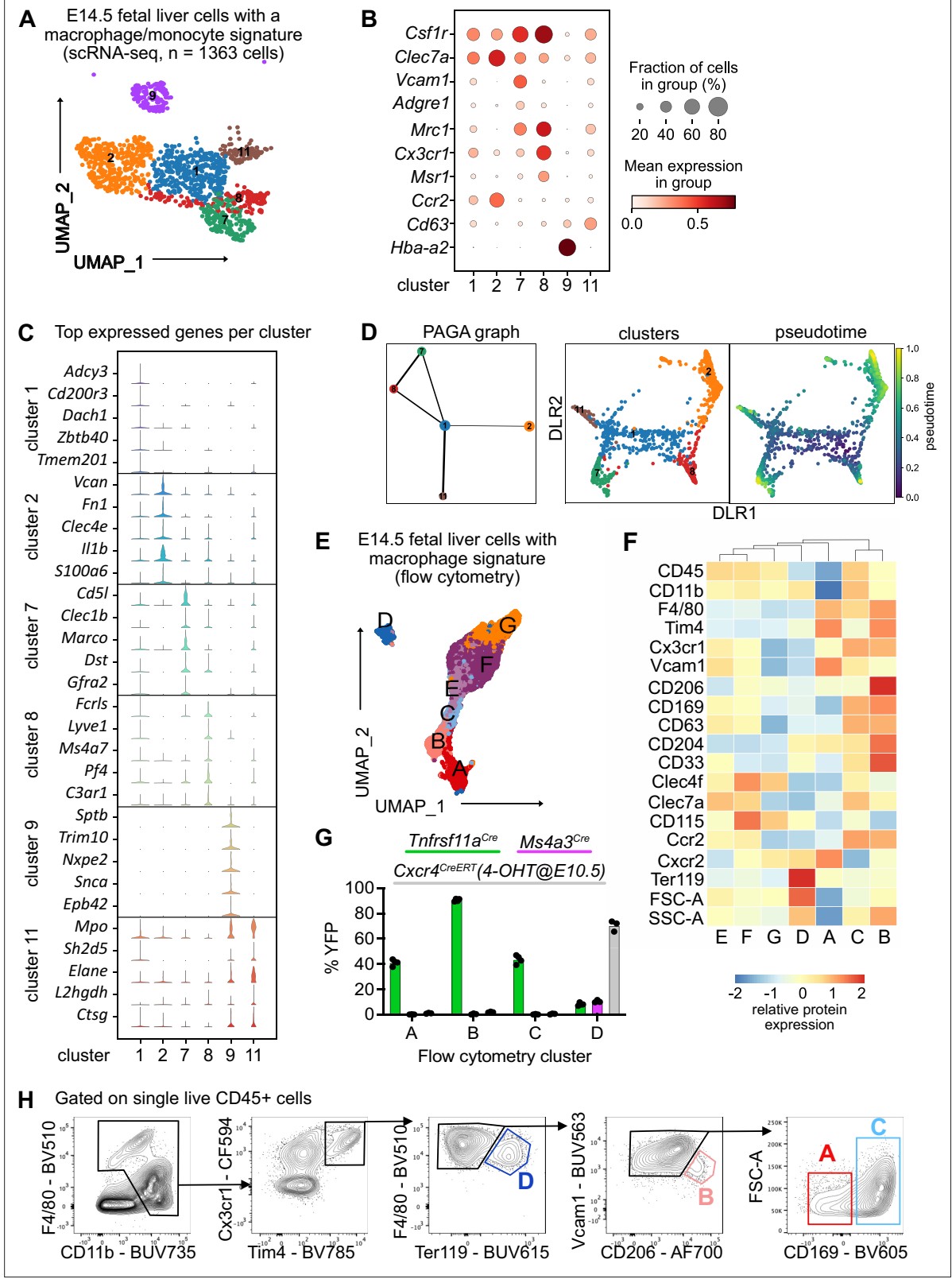

**Figure 1.** Characterization of fetal liver macrophage populations. (**A**) Single-cell RNA-sequencing (scRNA-seq) analysis of wildtype CD11b[low/+] cells isolated from a fetal liver at developmental day (E)14.5. Clusters of possible macrophage subsets identified by a monocyte/macrophage signature (see *Figure 1—figure supplement 1*) are visualized through Uniform Manifold Approximation and Projection (UMAP). (**B**) Expression of selected macrophage- and macrophage progenitor-specific genes in clusters from (**A**). (**C**) Violin plots of highly expressed genes within the clusters from (**A**). (**D**)

*Figure 1 continued on next page*

*Figure 1 continued*

Developmental trajectory analysis using Partition-based graph abstraction (PAGA) method (left) and pseudotime analysis (right) of the identified clusters in (**A**), excluding cluster 9. (**E**) Flow-cytometry analysis of CD11b$^{low/+}$ cells with a macrophage signature, isolated from a fetal liver at developmental day E14.5. Cell surface marker expression was used to generate unbiased clusters using UMAP. (**F**) Heatmap of relative protein expression and cell size parameters used in the flow-cytometry analysis in (**E**). (**G**) Flow-cytometry analysis of E14.5 fetal liver macrophages using three different fate-mapping mouse models. CD11b$^{low/+}$ cells with a macrophage signature were analyzed as shown in (**E, F**), resulting in similar cluster distribution (see *Figure 1— figure supplement 2*). YFP$^+$ cells from the *Tnfrsf11a$^{Cre}$; Rosa26$^{YFP}$* model (green) indicate a yolk sac origin. *Ms4a3$^{Cre}$; Rosa26$^{YFP}$* (pink) and *Cxcr4$^{CreERT}$; Rosa26$^{YFP}$* with 4-hydroxytamoxifen (4-OHT) injection at E10.5 (gray) indicate a monocytic and hematopoietic stem cell origin, respectively. Circles represent individual mice. (**H**) Simplified gating strategy to identify macrophage clusters in E14.5 livers using flow cytometry.

The online version of this article includes the following source data and figure supplement(s) for figure 1:

**Source data 1.** Quantification of fate-mapped macrophages populations for *Figure 1*.

**Figure supplement 1.** Sorting and characterization of fetal liver myeloid cells.

**Figure supplement 2.** Flow-cytometry analysis of macrophage subpopulations and fate-mapping mouse models.

Cluster 9 expressed almost exclusively erythroblast-specific genes (*Hba-a2*, *Sptb*, *Trim10*, *Nxpe2*, *Snca*, *Epb42*) (*Figure 1B,C*) and was identified as EI macrophages, which have been recently described as doublets of erythroblasts and macrophages (*Popescu et al., 2019*) or erythroblasts with cell remnants on their surface since macrophages frequently adhere to other cells (*Millard et al., 2021*). Clusters 7 and 8 showed the highest expression of bona fide macrophage genes such as *Csf1r*, *Mrc1*, and *Timd4* but were distinct in their expression of other macrophage markers (e.g., *Marco*, *Lyve1*) (*Figure 1B, C*, *Figure 1—figure supplement 1E*). Cells in cluster 2 expressed the highest levels of *Ccr2*, *Clec7a*, *Clec4a*, *Il1b*, and *S100a6* (*Figure 1B, C*) besides some of the macrophage-specific genes, hinting toward an inflammatory state of this putative macrophage population. Clusters 1 and 11 showed a low expression of core macrophage genes compared to clusters 2, 7, and 8, suggesting a precursor stage (*Figure 1B*). However, cluster 11 was distinct from cluster 1 with a high expression of granule-related genes, such as *Mpo*, *Elane*, and *Ctsg* (*Figure 1C*), which may indicate a granulo-cytic rather than a macrophage precursor state. To test these hypotheses and predict developmental trajectories, we performed a Partition-based graph abstraction (PAGA) analysis (*Wolf et al., 2019*) after excluding cluster 9 due to their doublet identity. Here, cluster 1 expression represents a progenitor state, thus, is the center of the network, which was confirmed by pseudotime analysis (*Figure 1D*). The other clusters fall into distinct nodes, with clusters 7 and 8 showing substantial similarity as indicated by the edge thickness (*Figure 1D*). In summary, our scRNA-seq analysis indicates the presence of at least three macrophage states (clusters 2, 7, and 8), which have distinct phenotypes.

To validate macrophage heterogeneity on the protein level, we performed a high-dimensional flow-cytometry analysis on CD11b$^{low/+}$ cells (*Figure 1—figure supplement 2A*). Similar to the scRNA-seq enrichment, we visualized all myeloid cells using UMAP, clustered them, and overlaid a macrophage signature (F4/80, Tim4, Cx3cr1, Vcam1, CD169, CD206, *Figure 1—figure supplement 2B*). This resulted in seven clusters, which we analyzed further (*Figure 1E*). Hierarchical clustering of all clusters expressing macrophage and macrophage precursor proteins confirmed the presence of three F4/80$^{high}$ macrophage populations (clusters A–C, *Figure 1F*). As already observed on the transcriptional level, cluster D likely represents Ter119$^+$ erythroblasts with macrophage cell remnants, as indicated by the increased cell size and granularity determined via forward scatter (FSC) and side scatter (SSC), respectively (*Figure 1F*). To correlate precursor and macrophage clusters identified by transcriptional and protein analyses, a correlation matrix between the scRNA-seq and flow-cytometry datasets was calculated based on gene expression corresponding to the presence of the respective antigen used in our flow-cytometry panel (*Figure 1—figure supplement 2C*). Here, scRNA-seq precursor clusters 1 and 11 corresponded highly to clusters F and G, which expressed high levels of CD45, CD115, and Clec4f but were low in F4/80 and Tim4 expression. In contrast, scRNA-seq clusters 2 and 7 represented cluster A while cluster 8 correlated mostly with clusters B and C, supporting the notion of three distinct macrophage populations in the fetal liver at E14.5, in addition to EI macrophages.

## Fetal liver macrophages originate from yolk sac progenitors

Next, we addressed the ontogeny of the macrophage clusters using *Rosa26$^{LSL-YFP}$* mice crossed to *Tnfrsf11a$^{Cre}$* for detection of pMac-derived cells (*Mass et al., 2016*), to *Ms4a3$^{Cre}$* for detection of monocyte-derived cells (*Liu et al., 2019*), and to the inducible *Cxcr4$^{CreERT}$* with 4-hydroxytamoxifen

injection at E10.5 labeling all cells of the definitive hematopoiesis wave (*Werner et al., 2020*). We confirmed the presence of all three macrophage clusters and the Ter119+ EI macrophage cluster in all mouse models (*Figure 1—figure supplement 2D*). HSC-derived definitive erythroblasts in cluster D were efficiently fate-mapped using the *Cxcr4CreERT* model, validating that these were cell doublets or erythroblasts with cell remnants on their surface since they also showed low labeling of the other fate-mapping models that label macrophages/monocytes (*Figure 1G*). The remaining clusters were YFP+ only in the *Tnfrsf11aCre* model, demonstrating that all fetal liver macrophages at E14.5 derive from pMacs. Using the distinct expression of Vcam1, CD206, and CD169 in clusters A–C and their difference in cell size allowed us to develop a simple gating strategy to distinguish these macrophage populations (*Figure 1H*). In summary, using a hypothesis-driven analysis of CD11blow/+ cells, we define, in addition to the already well-known EI macrophage population, distinct macrophage populations in the fetal liver that are yolk sac derived.

## Fetal liver macrophage subpopulations display distinct transcriptional programs

Next, we set out to investigate whether the macrophage heterogeneity we defined at E14.5 could indicate additional macrophage states besides serving as EI macrophages, therefore, resulting in other

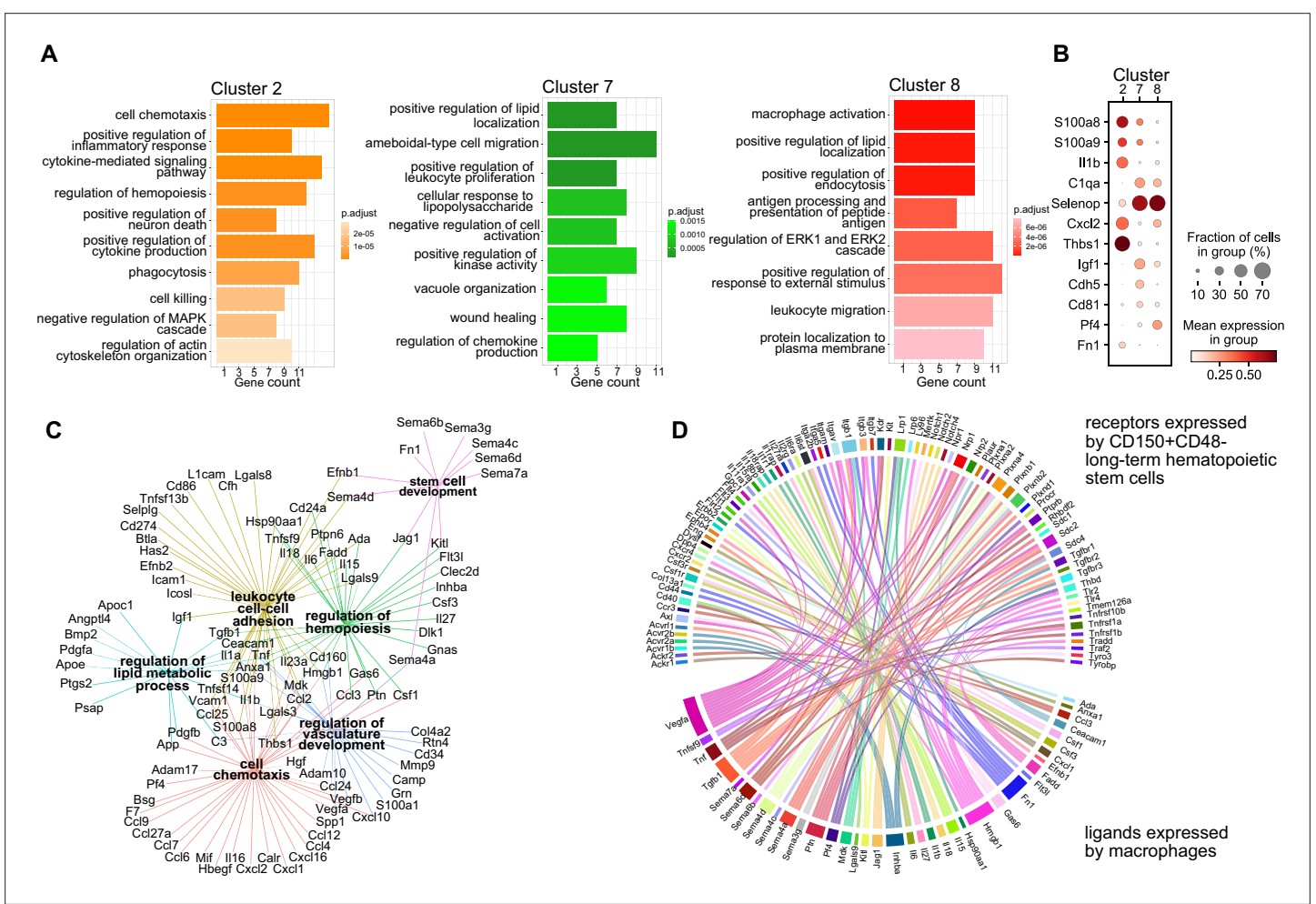

**Figure 2.** Transcriptional program and paracrine signaling of fetal liver macrophages. (**A**) Gene set enrichment analysis (GSEA) of final macrophage clusters 2, 7, and 8 was performed on the differentially expressed genes of each cluster. (**B**) Expression of selected ligands in the final macrophage clusters. (**C**) Interaction and functional network of expressed ligands on the identified macrophage clusters 2, 7, and 8. Each hub with a color indicates the function of the ligands, while the edges show the interaction between them. (**D**) Potential ligand–receptor interactions between macrophages and long-term hematopoietic stem cells (LT-HSCs). The gene names on the bottom of the plot are expressed ligands in macrophage clusters 2, 7, and 8. Gene names on the top are expressed receptors on LT-HSCs at E14.5. Each ligand can target several receptors which are indicated with the same color.

functions than phagocytosing erythroblast nuclei. To this end, we first performed a gene ontology (GO) term analysis on the top expressed 100 genes, comparing each cluster from the scRNA-seq analysis to all other clusters. Here, cluster 2-specific genes fell into the terms 'cell chemotaxis', 'positive regulation of inflammatory response', and 'cytokine-mediated signaling pathway' (*Figure 2A*), indicating the activated inflammatory state already observed in the top 5 expressed genes (*Figure 1C*). In contrast, cluster 7-related genes were significant for the GO terms 'positive regulation of lipid localization', 'ameboidal-type cell migration', and 'apoptotic cell clearance'. Also cluster 8 expressed genes belonging to the GO term 'positive regulation of lipid localization' that partially overlapped with cluster 7 (e.g., *Apoe*, *Lpl*) (*Supplementary file 1*). Additional cluster 8-specific terms were 'macrophage activation' and 'cell junction disassembly' (*Figure 2A*). This analysis indicated that, albeit somewhat similar, the distinct macrophage populations might exert distinct functions in the fetal liver. This was confirmed by the cluster-specific expression of selected ligands indicating that the distinct cellular states may result in distinct paracrine signaling activity (*Figure 2B*). Intersecting the CellTalk database information (*Shao et al., 2021*) with the complete set of genes expressed by the three macrophage states revealed 208 potential macrophage-derived secreted ligands (*Supplementary file 2*).

We next used the 208 ligand candidates and performed a gene set enrichment analysis, and visualized the gene/GO term relationships in a network (*Figure 2C*). This analysis pointed to additional functionality of the three macrophage states beyond erythropoiesis, which included regulation of hemopoiesis and stem cell development together with chemotaxis and vasculature development, mechanisms that could shape the stem cell niche.

Macrophages contribute to the stem cell niche, particularly in the bone marrow, under inflammatory conditions (*Seyfried et al., 2020*). Yet, evidence for macrophage-derived molecules involved in the direct cell crosstalk controlling stem cell maintenance and differentiation in the mouse fetal liver is missing. Therefore, we sought to explore potential signaling events between macrophages and HSCs to determine whether macrophage-derived factors might modify HSC function during steady state. Thus, we sequenced LT-HSC at E14.5 and, leveraging our scRNA-seq macrophage dataset (see Methods), uncovered potential ligand–receptor interactions between macrophage-derived ligands and LT-HSCs in the fetal liver (*Figure 2D*). Many of the ligands are well-known players in the stem cell niche, for example, Kitl, Igf1, Tnf, Tgfb1, and Fn1, which have been reported to directly or indirectly promote the expansion of HSPCs (*Azzoni et al., 2018*; *Hadland et al., 2022*; *Sakaki-Yumoto et al., 2013*). In summary, our transcriptomic ligand–receptor interaction analyses suggest that macrophages express HSC niche factors and, thereby, may actively contribute to LT-HSC functionality in the fetal liver.

## Macrophages interact directly with LT-HSCs

We next asked whether the potential paracrine signaling between macrophages and LT-HSCs in the fetal liver occurs via direct interaction. To test this, we first performed immunofluorescence analyses on E14.5 liver cryosections by staining macrophages with Iba1 and F4/80 and LT-HSCs with CD150, indicating a direct interaction between these cells (*Figure 3A*). Next, we performed stainings on 3D whole-mount livers at E14.5 using different macrophage-, stem cell-, and erythroblast-specific markers. We found EI macrophages that were entirely surrounded by Ter119+ erythroblasts, as expected, but frequently observed cell–cell interactions of macrophages with c-Kit+ progenitors and CD150+ LT-HSCs (*Figure 3B*). 3D reconstruction analysis of Iba1+ macrophage morphology in combination with CD150+ and CD150− cell positioning (*Figure 3C*) showed that CD150+ cells were clearly attracted to macrophage surfaces, as indicated by an increased density within a distance of 3 μm compared to random CD150− cells (*Figure 3D*). Quantification of the average distance to an Iba1+ cell demonstrated that CD150+ cells exhibit significantly closer proximity to macrophages compared to CD150− cells, suggesting a preferential positioning of CD150+ cells nearby macrophages (*Figure 3E*).

To determine whether LT-HSCs interact with distinct macrophage populations, we assessed cellular interactions via co-detection by indexing (CODEX)-enabled high-dimensional imaging (*Black et al., 2021*; *Frede et al., 2022*; *Goltsev et al., 2018*; *Figure 4A*). CD45, Iba1, F4/80, Cx3cr1, and Tim4 were used as pan-macrophage markers, allowing the distinction of cluster A: CD106 (Vcam1)+ macrophages, cluster B: CD206+ macrophages, cluster C: CD169+ macrophages, and the most abundant cluster D: Ter119+ EI macrophages (*Figure 4B*, *Figure 4—figure supplement 1A*). First, we assessed the distribution of the four macrophage clusters in the different liver lobes using a Voronoi diagram,

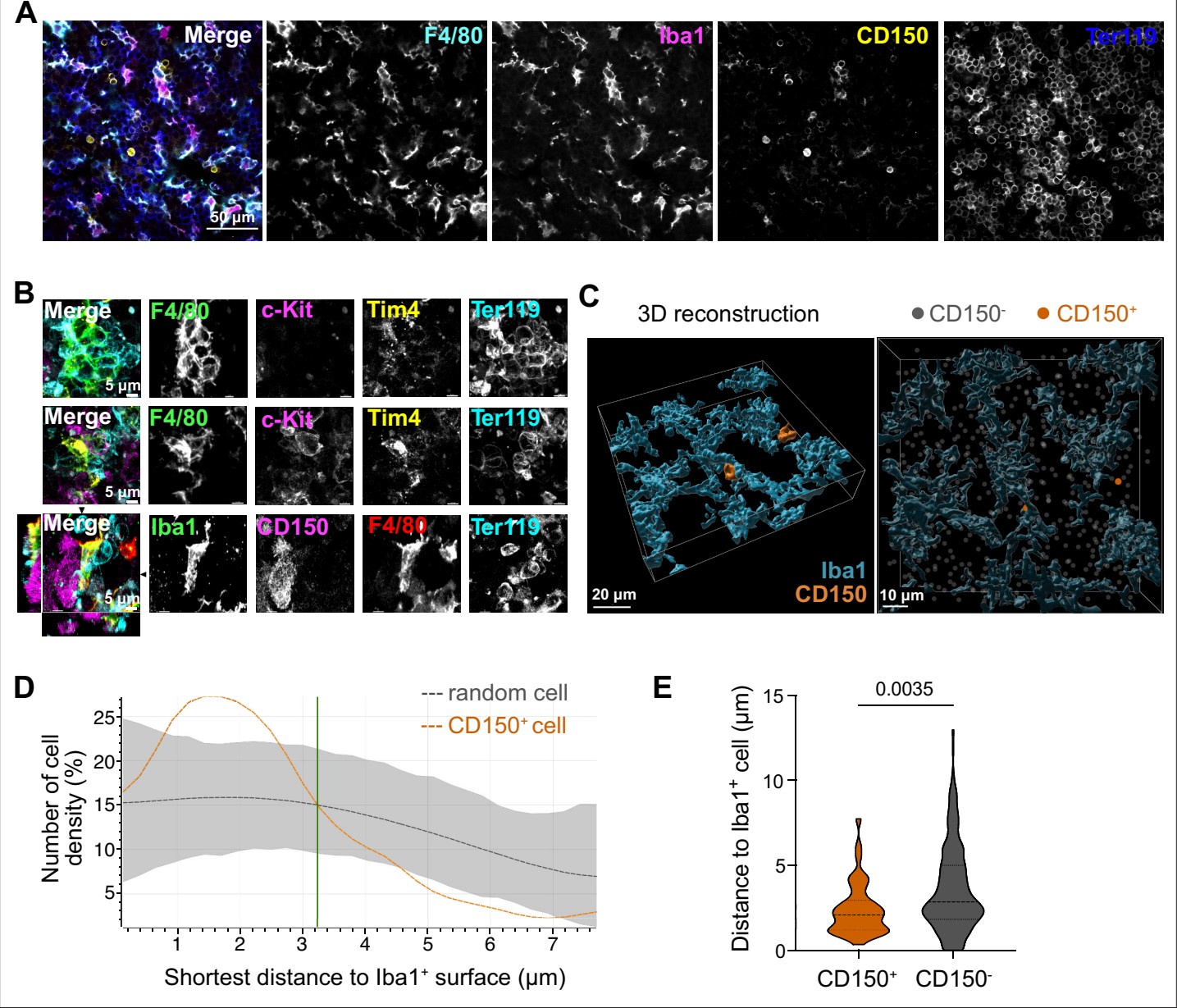

**Figure 3.** Macrophage interaction with CD150+ hematopoietic stem cells (HSCs). (**A**) Immunostaining of E14.5 fetal liver sections with antibodies against F4/80, Iba1, CD150, and Ter119. Scale bar represents 50 μm. (**B**) Immunostaining of E14.5 fetal liver whole-mounts with antibodies against F4/80, c-Kit, Tim4, Ter119, Iba1, and CD150. The first row visualizes the interaction between EI macrophages surrounded by Ter119+ erythroblasts. The second and last rows highlight the interaction between macrophages and progenitor cells, including CD150+ long-term HSCs (LT-HSCs). The black arrows on the last row indicate the *yz* and *xz* dimensions which are shown on the left and bottom sides. Scale bar represents 5 μm. (**C**) 3D reconstruction of a E14.5 fetal liver whole mount stained with Iba1, CD150, and DAPI (shown as gray dots indicating CD150- cells). Scale bar represents 20 μm (left) or 10 μm (right). Measured distance of CD150+ and CD150− cells to the next Iba1+ macrophage (**D**) and average distance of cells to Iba1+ macrophages (*n* = 42 for CD150+ and *n* = 524 for CD150−) (**E**). Unpaired Student's *t*-test.

The online version of this article includes the following source data for figure 3:

**Source data 1.** Distances measured between CD150+ or CD150− cells and Iba1 cells for *Figure 3*.

showing that all macrophage clusters are found across the whole tissue (*Figure 4C*). CD150+ HSC were also dispersed across the fetal liver but showed a preferential localization near the liver capsule (*Figure 4C*). Similar results were detected via spatially segmented cellular neighborhoods of the single objects using a raster scan with a radius of 50 μm and a self-organizing map (SOM) algorithm

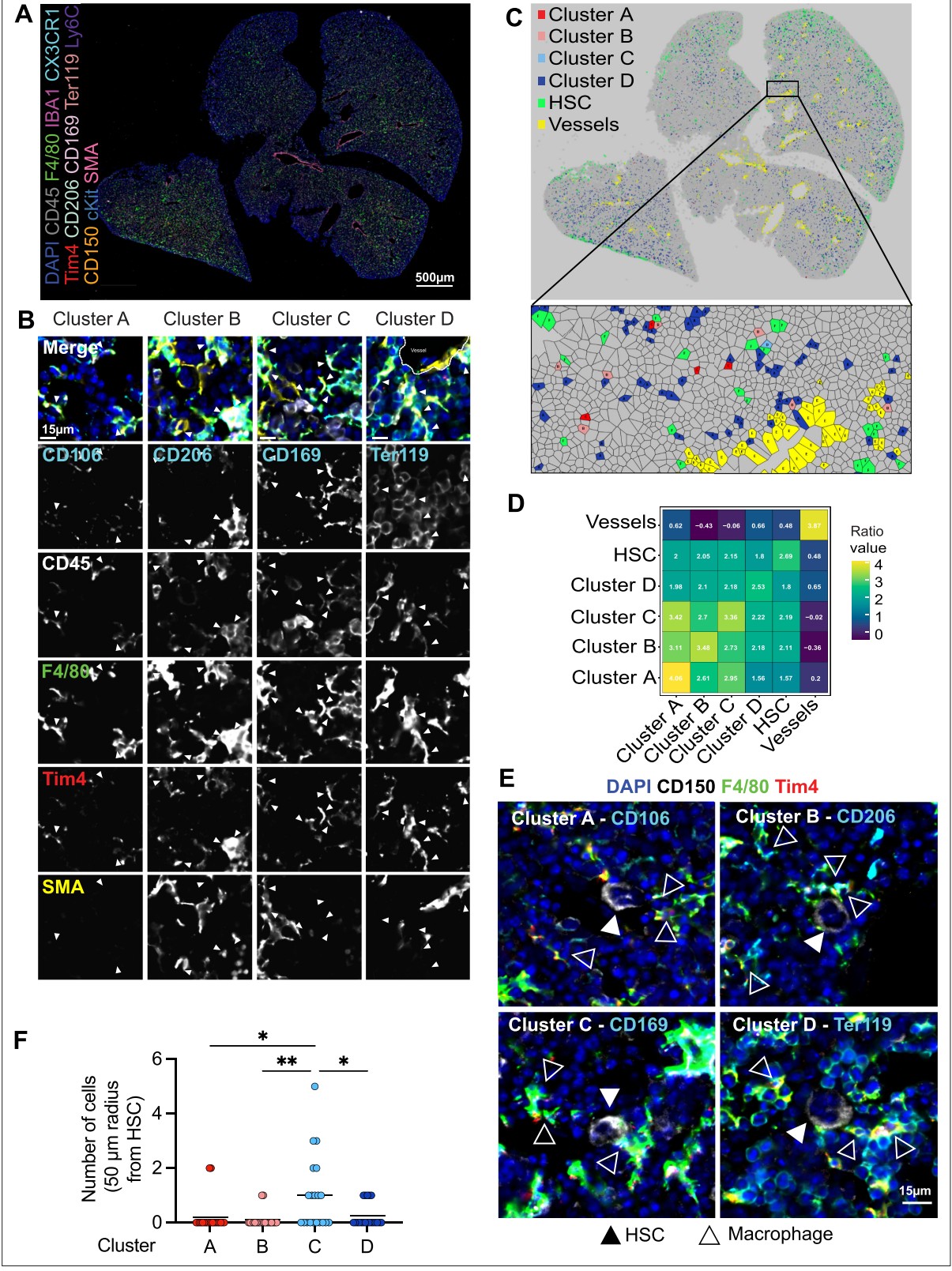

**Figure 4.** Macrophage heterogeneity and their interaction with hematopoietic stem cells (HSCs). (**A, B**) A 5-μm frozen section of a fetal liver from an E14.5 wildtype embryo was stained with a 20-plex CODEX antibody panel. Representative image of the entire field of view is shown in (**A**) and enlargements showing representative images of the cells from clusters A–D are shown in (**B**). Arrows in the enlargements indicate macrophages from the corresponding cluster. Scale bars represent 500 μm in the overlay and 15 μm in the enlargements. (**C**) Voronoi diagram from (**B**) after manual cell

*Figure 4 continued*

classification using the HSCs, blood vessels, and cells from the four macrophage clusters as seeds. (**D**) Spatial analysis of interactions between cells from macrophage clusters, HSCs, and blood vessels in the fetal liver within a range of 5–50 µm. Values represent the calculated Log Odds Ratio. (**E**) Representative images of the interaction of HSCs with the macrophage clusters. Filled arrowheads indicate the HSC, and empty arrowheads indicate the macrophages from the corresponding cluster. CD150 staining is shown in white color. Scale bar represents 15 µm. (**F**) Absolute number of cells from the four macrophage clusters within a radius of 50 µm from one randomly selected HSC. Each dot represents one cell. Black lines in the plot represent the mean. One-way analysis of variance (ANOVA); *p < 0.05, **p < 0.01.

The online version of this article includes the following source data and figure supplement(s) for figure 4:

**Source data 1.** Number of CD150$^+$ cells within each macrophage subpopulation measured in a 50-µm radius.

**Figure supplement 1.** Spatial CODEX analyses of macrophage clusters in the fetal liver.

**Figure supplement 1—source data 1.** Quantification of macrophage subpopulations and their interaction with Ter119$^+$ cells for *Figure 4—figure supplement 1*.

**Figure supplement 2.** CODEX-based detection of macrophage heterogeneity and hematopoietic stem cells (HSCs).

**Figure supplement 2—source data 1.** Quantification of CD150 and CD41 expression for *Figure 4—figure supplement 2*.

where CD150$^+$ HSCs, represented by neighborhood 4, were mainly found near the liver capsules of the lower left and the two upper liver lobes (*Figure 4—figure supplement 1B*). Next, we used a data-driven approach to detect spatial interactions between macrophage clusters and HSCs within a range of 5–50 µm. Interestingly, LT-HSCs showed the highest correlation with macrophages from cluster C representing CD169$^+$ macrophages (*Figure 3D*) and the lowest correlation with cluster D (Ter119$^+$ macrophages). Manual inspection of cells surrounding CD150$^+$ HSC in a 50-µm radius also revealed that LT-HSCs were preferentially surrounded by CD169$^+$ macrophages (cluster C, *Figure 3E*), with fewer identified cells belonging to the other macrophage populations (*Figure 3F*).

As erythroblasts are the most abundant cell type, we asked whether clusters A–C serve, at least partially, as EI macrophages. To this end, 50 macrophages of each cluster were randomly chosen, and the direct interaction with Ter119$^+$ cells was evaluated manually. While cells belonging to cluster D (Ter119$^+$ EI macrophages) showed 100% interaction, as expected, cluster A (CD106$^+$ macrophages, 73% interaction), cluster B (CD206$^+$ macrophages, 80% interaction), and cluster C (CD169$^+$ macrophages, 80.4% interaction) did not always interact with Ter119$^+$ erythroblasts (*Figure 4—figure supplement 1C, D*). The CD206$^+$ cluster A was the subpopulation with the least interaction and longest distance to the nearest erythroblast (*Figure 4—figure supplement 1E*), which was often accompanied by an elongated cell shape near vessels, indicating the presence of CD206$^+$ perivascular macrophages (*Figure 4—figure supplement 2A, B*). Pairwise overlay of specific macrophage cluster markers revealed that there is partial overlap of the macrophage subclusters, with the highest overlap of CD206 and CD169 expression (Pearson correlation = 0.295) and the lowest overlap of CD106/Vcam-1 (Pearson correlation = 0.081) (*Figure 4—figure supplement 2C*), validating results observed via flow cytometry (*Figure 1F*).

Finally, we made use of the fate-mapping model *Cxcr4$^{CreERT}$; Rosa26$^{LSL-YFP}$* to validate that CD150$^+$ cells at E14.5 are majorly Cxcr4$^+$ LT-HSCs and not CD41$^+$ megakaryocyte progenitors that express CD150 in the adult bone marrow (*Pronk et al., 2007*). Using CODEX-based gating of CD150$^+$ cells, we detected no overlap with CD41 protein expression (*Figure 4—figure supplement 2D, E*). This is in line with previously published data where CD150$^+$ LT-HSCs were sorted from E15 livers (*Ghosn et al., 2016*). Furthermore, we observed GFP/YFP expression on CD150$^+$ CD41$^-$ cells (*Figure 4—figure supplement 2D*), which indicates that these cells are either currently expressing Cxcr4 and/or have been labeled at an earlier developmental day by the *Cxcr4$^{CreERT}$; Rosa26$^{LSL-YFP}$* model, confirming that this model traces definitive hematopoiesis when 4-hydroxytamoxifen is injected at E10.5 or later developmental time points.

Of note, the tissue analyzed via CODEX represents only cellular neighborhoods in X and Y due to the thin sectioning technique (5 µm) and, thus, does not take neighboring cells in the Z plane into account. This leads to an underestimation of macrophage–HSC interactions in comparison to the 3D reconstruction analysis (*Figure 3C–E*). In summary, our data indicate that macrophages inhabit distinct niches within the fetal liver, with the majority of macrophages supporting mainly erythroblast maturation, while other populations may support HSC function.

## Lack of macrophages leads to decreased erythrocyte maturation

Given that LT-HSCs have direct contact with or are in close proximity to macrophages providing niche signals, we hypothesized that depletion of fetal liver macrophages would alter the LT-HSC phenotype and function. Therefore, we took advantage of the *Tnfrsf11a*<sup>Cre/+</sup>*; Spi1*<sup>f/f</sup> mouse model, which should lead to a depletion of fetal macrophages since *Tnfrsf11a* is expressed by pMacs (***Mass et al., 2016***) and *Spi1* (also known as *Pu.1*) is required for macrophage differentiation (***McKercher et al., 1996***; ***Scott et al., 1994***). Indeed, flow cytometry and immunostaining revealed an 80–90% reduction of F4/80+/Iba1+ cells in fetal livers of *Tnfrsf11a*<sup>Cre/+</sup>*; Spi1*<sup>f/f</sup> embryos (hereafter referred to as *KO*) compared to littermate controls (*WT*) demonstrating efficient depletion of macrophages (***Figure 5A, B***). First, we analyzed the absolute cell numbers and the number of CD45+ cells per fetal liver, to ensure that our depletion strategy did not have any major off-targets leading to a developmental delay. We did not observe any changes in cell numbers (***Figure 5C***) or tissue architecture (***Figure 5D***). Erythroblast maturation is characterized by changes in CD71 and Ter119 expression, allowing us to determine their developmental sequence within six subsets (from S0 to S5) (***Fraser et al., 2007***; ***Pop et al., 2010***). The final maturation step of enucleation to produce a functional erythrocyte depends on EI macrophages (***Palis, 2014***). Indeed, Giemsa staining of blood smear samples from *KO* embryos showed a reduction of enucleated erythrocytes compared to controls (***Figure 5E***). However, the maturation of erythroblasts in the liver was not altered (***Figure 5F***). These results suggest that our newly developed mouse model efficiently targets fetal liver macrophages at E14.5, leading to a delayed erythrocyte enucleation, consistent with the function of EI macrophages.

## Depletion of macrophages leads to transcriptional changes in HSCs

To determine whether the lack of macrophages would impact LT-HSC functionality, we performed bulk RNA-sequencing on sorted LT-HSCs from *KO* embryos and littermate controls at E14.5 (***Figure 6— figure supplement 1A***). Analysis of differentially expressed genes (DEGs) resulted in 598 upregulated and 555 downregulated genes (***Figure 6A***, ***Figure 6—figure supplement 1B***, ***Supplementary file 3***). Some of the upregulated genes were well-known transcriptional regulators of hematopoietic specification and stem cell capacity, such as *Gata2* and *Gata3* (***Figure 6A***, ***Figure 4—figure supplement 2C***). Examining GO pathways of these DEG revealed signaling mechanisms enriched for metabolic processes, organelle localization and RNA-related processes to be downregulated (***Figure 6B***, ***Supplementary file 4***). In contrast, genes belonging to the GO terms chromatin organization, myeloid cell differentiation, regulation of hemopoiesis, and mononuclear cell proliferation were upregulated (***Figure 6B***, ***Figure 6—figure supplement 1C***). These data indicate that the transcriptional program of LT-HSC is regulated by macrophages and, thereby, may impact their proliferative and/or differentiation potential.

## Depletion of macrophages does not change stem and progenitor cell numbers

To assess LT-HSC maintenance, retention and proliferation in the fetal liver, we first performed flow-cytometry experiments to quantify stem and progenitor cell numbers (gating strategy, ***Figure 7— figure supplement 1A***). Quantification of LT-HSCs, short-term (ST)-HSCs, MPP2, MPP3, and MPP4 did not reveal any significant differences between *KO* and littermate controls at E14.5 (***Figure 6C***). Further downstream progenitors, such as the common lymphoid progenitor, common myeloid progenitor, megakaryocyte–erythroid progenitor, and granulocyte–macrophage progenitor (GMP), were not significantly altered in cell numbers, albeit there was a tendency for increased GMP numbers in *KO* livers (***Figure 6D***). To address the proliferation capacity of HSCs, we sorted single LT-HSCs from *KO* and littermate controls and monitored their proliferation 48 hr later (***Figure 6E***). Furthermore, we performed serial colony-forming unit (CFU) assays to study the long-term self-renewal ability (***Figure 6F***). In both assays, *KO* stem cells showed no defects or increase in proliferation compared to littermate controls (***Figure 6E, F***). These results suggest that a reduction of macrophages in the HSC niche at E14.5 does not modify stem and progenitor cell numbers or lead to a dysregulated proliferation capacity of LT-HSCs.

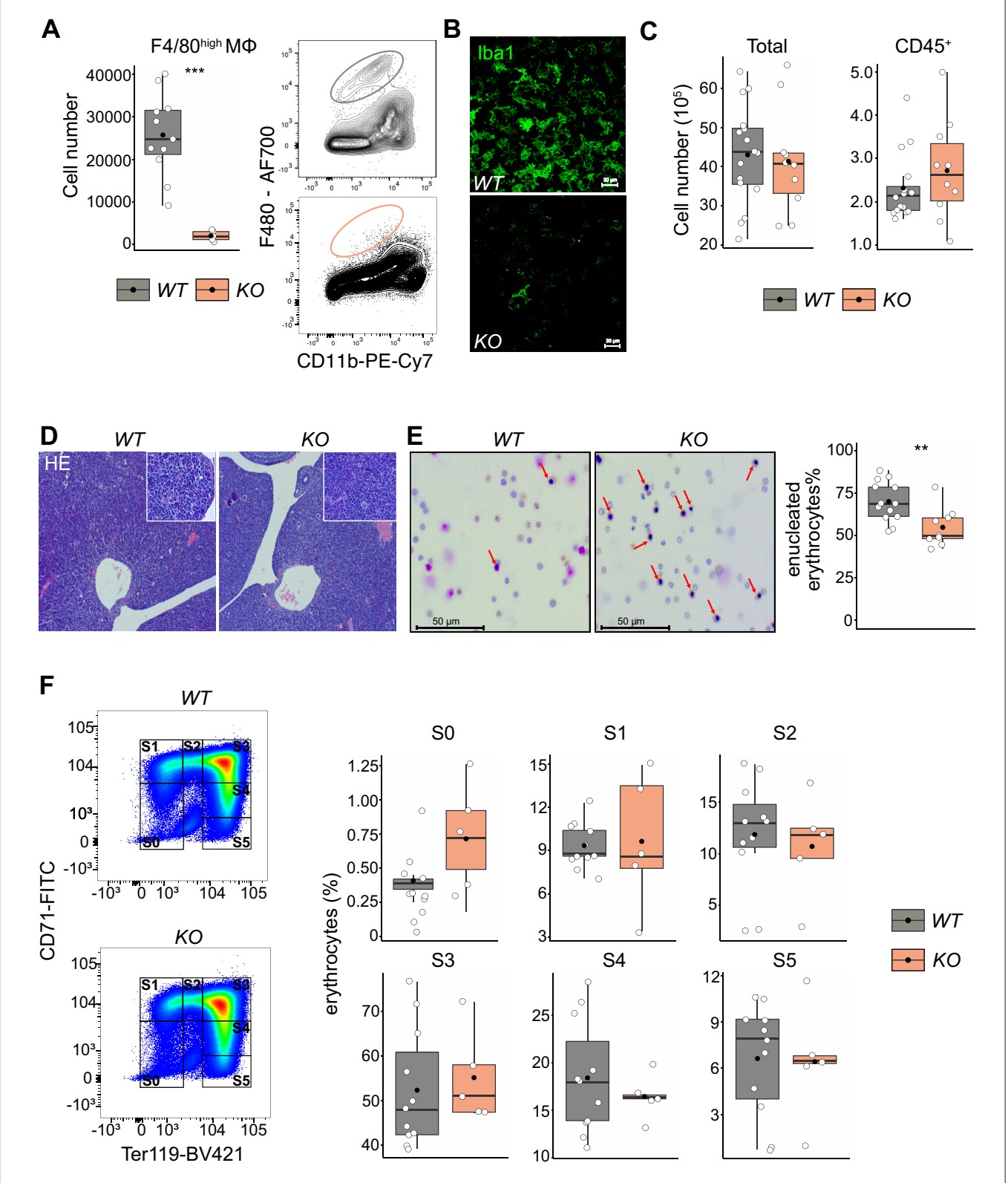

**Figure 5.** The effect of macrophage depletion on erythropoiesis. (**A**) Quantification of F4/80+ macrophage cells in the E14.5 fetal liver of *Tnfrsf11a+/+; Spi1f/f* (*WT*) and *Tnfrsf11aCre/+; Spi1f/f* (*KO*) embryos using flow cytometry. *n* = 16 for *WT*, *n* = 10 for *KO*. (**B**) Immunofluorescent staining of macrophages using Iba1 antibody in E14.5 livers. Representative for *n* = 3. Scale bar represents 30 μm. (**C**) Quantification of single live and CD45+ cells in the E14.5 fetal liver of *WT* and *KO* knockout embryos using flow cytometry. *n* = 11 for *WT*, *n* = 5 for *KO*. (**D**) Hematoxylin and eosin stain (HE) of *WT* and *KO* fetal

*Figure 5 continued on next page*

*Figure 5 continued*

livers at E14.5. Representative for *n* = 5. Overviews were taken with a ×5 objective, insets with a ×20 objective. (**E**) On the left: representative pictures of blood smear using May–Grünwald–Giemsa staining. Arrows indicate nucleated erythroblasts. On the right; the percentage of enucleated erythrocytes in the blood of *WT* and *KO* embryos at E14.5. *n* = 13 for *WT* and *n* = 9 for *KO*. (**F**) On the left: representative gating strategy to capture differentiation stages of erythrocytes. On the right: comparison of the erythrocyte's percentages in each of the differentiation stages between *WT* and *KO* embryos at E14.5. *n* = 11 for *WT* and *n* = 5 for *KO*. All statistical tests comparing *WT* and *KO* embryos: ***p < 0.001, **p < 0.01, Wilcoxon test.

The online version of this article includes the following source data for figure 5:

**Source data 1.** Quantified cell numbers of knockout and wildtype E14.5 embryos for **Figure 5**.

## Macrophages control HSC differentiation potential

Next, we addressed the differentiation behavior of HSCs in vivo and in vitro. Using flow cytometry, we focused on myeloid cells since genes important for myeloid cell differentiation were upregulated (**Figure 6B**, **Figure 6—figure supplement 1C**). First, we assessed the relative proportions of macrophages, monocytes, and neutrophils in *KO* and littermate controls at E14.5 (**Figure 7A**), indicating that the loss of macrophages leads to a relative expansion of both neutrophils and monocytes. An unbiased clustering of cells using UMAP indicated a reduction of F4/80$^+$ macrophage clusters A–C, but not of cluster D (**Figure 7B**), which again underlines the fact that these F4/80$^+$Ter119$^+$ events represent cell doublets or erythroblasts with attached macrophage cell remnants. Quantification of total numbers showed that macrophage subclusters A–C were significantly reduced in *KO* livers (**Figure 7C**).

In addition to a Ly6G$^+$ neutrophil cluster in the UMAP, we detected two Ly6C$^+$ monocyte clusters that were distinguished by their Cx3cr1 expression (**Figure 7B**). Using a gating strategy to detect these myeloid cell types (**Figure 7—figure supplement 1B**), we observed a significant increase of Ly6G$^+$ cells in *KO* livers compared to littermate controls while the number of monocytes was not altered (**Figure 7D**). Intriguingly, neutrophils numbers remained high in *KO* livers throughout embryonic development compared to littermates, albeit the empty macrophage niche was replenished by E16.5 and remained stable at E18.5 (**Figure 7E, F**). To test whether the change in differentiation potential of *KO* HSCs is cell autonomous, we first performed a CFU-C assay. While the total number of colonies was similar in both genotypes, the number of CFU-granulocyte/macrophage (GM) was increased (**Figure 7G**). To test the hypothesis of a cell-autonomous effect in vivo, we made use of an adoptive transfer model where we transplanted either *KO* or *WT* fetal liver cells into lethally irradiated CD45.1 hosts (**Figure 7H**). The chimerism was almost at 100% CD45.2 after 12 weeks indicating that HSCs of the fetal liver have completely taken over hematopoiesis of the adult host. Similar to results observed in the developing embryo and CFU assays, *KO* HSCs produced significantly more neutrophils compared to littermate control HSCs (**Figure 7I**). As neutrophils originate from GMPs, which also give rise to monocytes and which are slightly increased in the E14.5 liver (**Figure 6D**), we additionally quantified Ly6C$^{high}$ monocytes in the blood of the host. Similarly to Ly6G$^+$ neutrophils, monocytes were significantly increased in numbers in hosts harboring HSCs from KO fetal livers (**Figure 7I**). Together with the results from the RNA-seq analyses, our data indicate that the lack of fetal liver macrophages causes a reprogramming of LT-HSCs, leading to their preferential differentiation toward the GMP-dependent lineage in vitro and in vivo.

## Discussion

We have shown that liver macrophages at E14.5 are heterogenous and that they play an active role in the niche of CD150$^+$ LT-HSCs. While most macrophages serve purely as EI macrophages, being only surrounded by Ter119$^+$ erythroblasts and promoting erythroblast maturation, a subset of macrophages directly interacts with other cells, such as c-Kit$^+$ and CD150$^+$ stem and progenitor cells. Being part of the HSCs niche, macrophages seem to specifically control the production of neutrophils, likely via paracrine factors that imprint the tissue environment on LT-HSCs, thereby enabling a tight balance of hematopoietic cell numbers. The increased monocyte production originating from transplanted *KO* HSCs suggests that the presence of macrophages may contribute specifically to the differentiation trajectory from LT-HSCs toward GMPs. The unchanged numbers of monocytes in the developing embryo are likely due to the requirement of fetal monocytes to quickly differentiate into macrophages that replenish the empty niche. Using various fate-mapping models, we could exclude any contribution

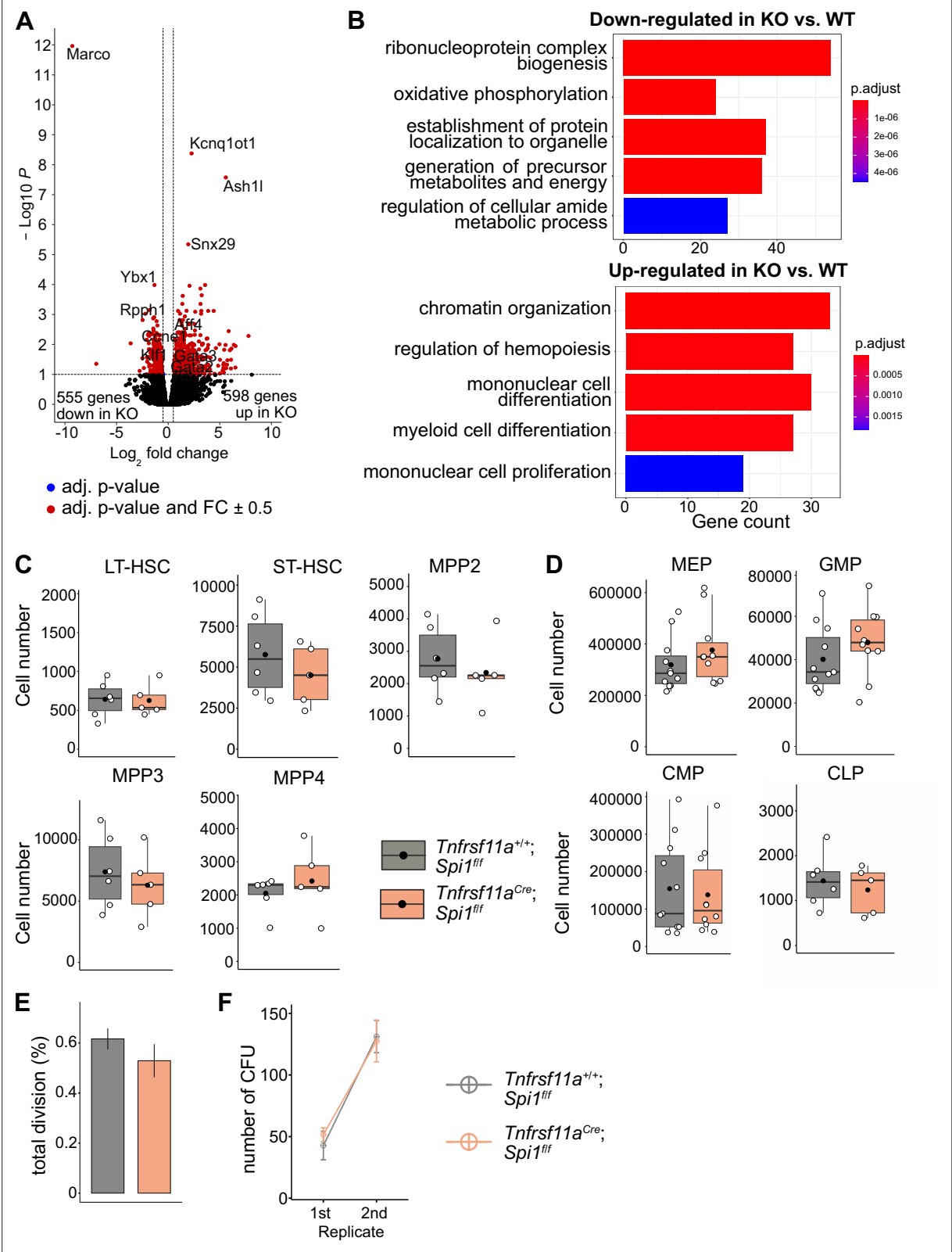

**Figure 6.** The effect of macrophage depletion on hematopoiesis. (**A**) Volcano plot of differentially expressed genes between CD150+ LT-HSCs sorted from *Tnfrsf11a+/+; Spi1f/f* (*WT, n = 4*) and *Tnfrsf11aCre/+; Spi1f/f* (*KO, n = 6*) embryos. Blue dots are significant (adjusted p-value <0.1), red dots are significant with a fold-change of ±0.5. (**B**) Gene set enrichment analysis of significant up- and downregulated genes from (**A**) between the control and knockout embryos. (**C**) Quantification of the stem- and progenitor cells from *WT* and *KO* fetal livers at E14.5. LT-HSC: long-term hematopoietic stem

*Figure 6 continued on next page*

*Figure 6 continued*

cells; ST-HSC: short-term hematopoietic stem cells; MPP: multipotent progenitors. *n* = 6 for *WT* and *n* = 5 for *KO*. (**D**) Quantification of progenitors from *WT* and *KO* fetal livers at E14.5. *n* = 6–10 for *WT* and *n* = 5–10 for *KO*. CLP: common lymphoid progenitor; CMP: common myeloid progenitor; GMP: granulocyte–macrophage progenitor; MEP: megakaryocyte–erythrocyte progenitor. (**E**) Cell proliferation assay. LT-HSCs were harvested from the fetal liver at E14.5 using FACS for performing a single-cell colony assay. *n* = 77 LT-HSCs from *n* = 5 fetal livers for *WT* and *n* = 96 LT-HSCs from *n* = 5 fetal livers for *KO*. (**F**) Serial transfer colony-forming assay showing the number of observed colonies after seeding E14.5 fetal liver cells into media (1st replicate) and re-seeding the cultured colonies (2nd replicate). *n* = 7 for *WT* and *n* = 8 for *KO*.

The online version of this article includes the following source data and figure supplement(s) for figure 6:

**Source data 1.** Quantified cell numbers of knockout and wildtype E14.5 livers and cells from serial colony-forming unit (CFU) and proliferation assays for *Figure 6*.

**Figure supplement 1.** Bulk RNA-sequencing and analysis of long-term hematopoietic stem cell (LT-HSC).

of definitive HSCs to the fetal liver macrophage pool. Finally, we provide a simple gating strategy with commonly available antibodies that allow the identification of macrophage subpopulations and the discrimination from cell doublets.

CODEX analyses indicate that CD150$^+$ LT-HSCs are preferentially found in close proximity to CD169$^+$ macrophages. CD169, also known as sialoadhesin, is a cell adhesion protein that has been described as an EI macrophage marker in the bone marrow and the fetal liver (*Chow et al., 2013*; *Chow et al., 2011*; *Li et al., 2019*; *Seu et al., 2017*), albeit with varying expression patterns (*Seu et al., 2017*). In line with this, our flow-cytometry and in situ immunofluorescent CODEX analysis of the fetal liver indicates that the majority of macrophages express CD169, with few macrophages that are CD169-negative and small in size but that express high levels of F4/80, Tim4, and Vcam1, and can thus be considered *bona fide* macrophages (cluster A). CD169 has been further defined as essential for erythropoiesis by promoting erythroblast maturation in the bone marrow (*Chow et al., 2013*). Intriguingly, CD169 is not required to bind erythroblasts, as shown by studies using specific inhibitors (*Morris et al., 1991*; *Morris et al., 1988*), but instead accumulates in contact zones between macrophages and immature granulocytes (*Crocker et al., 1990*). However, these studies, as many others analyzing EI macrophages, were performed after flushing the bone marrow. Thus, an ultrastructural characterization of the fetal liver in situ may be helpful in addressing whether CD169 forms clusters in the plasma membrane, which may be a direct interaction zone between CD169$^+$ macrophages and LT-HSCs. However, since CD150$^+$ LT-HSCs also interacted with CD169$^-$ macrophages, the tethering mechanism may rely on another surface receptor altogether.

Recent studies have addressed the role of fetal macrophages in the development of HSPCs. Work in zebrafish shows a homing and retention mechanism controlled by macrophages (*Li et al., 2018*; *Theodore et al., 2017*). Li et al. describe a Vcam1$^+$ macrophage-like cell population that interacts with HSPCs and serves as a permissive signal for HSPC entry into the embryonic caudal hematopoietic tissue (CHT) niche (*Li et al., 2018*). Yet, our work does not support an essential role of macrophages for LT-HSC homing or retention to the fetal liver since numbers of LT-HSC are not affected in *Tnfrsf11a*$^{Cre/+}$; *Spi1*$^{f/f}$ embryos. Work in mouse embryos indicates that CD206$^+$ macrophages in the AGM contribute to intra-aortic HSC generation and maturation (*Mariani et al., 2019*). Furthermore, macrophages have been suggested to promote HSC/MPP proliferation in the fetal liver (*Gao et al., 2022*). However, due to the lack of genetic mouse models targeting only yolk sac-derived macrophages and not the definitive wave of hematopoiesis, these studies relied on macrophage depletion via clodronate liposomes and the Csf1r inhibitor BLZ945 (*Gao et al., 2022*; *Mariani et al., 2019*). Thus, the long-term impact of these substances on the proliferation capacity of HSCs or other niche cells that promote HSC development cannot be excluded. Indeed, our model, in which we target pMacs very efficiently using the *Tnfrsf11a*$^{Cre}$ mouse model (*Mass et al., 2016*), leading to an almost complete depletion of macrophages while HSCs remain wildtype, we do not observe any defects in the proliferation and expansion of HSCs arguing for an unspecific off-target effect of clodronate and BLZ945.

During steady-state adulthood, macrophages have been shown to control neutrophil numbers through the clearance of apoptotic neutrophils and via the G-CSF/IL-17/IL-23 cytokine axis, which promotes granulopoiesis (*Gordy et al., 2011*; *Hong et al., 2012*; *Stark et al., 2005*). The reduction of macrophage populations in the bone marrow and spleen observed in *Lyz2*$^{Cre}$; *Cflar*$^{f/f}$ mice led to neutrophilia during steady state, which was attributed to the defect in efferocytosis of apoptotic neutrophils (*Gordy et al., 2011*). Furthermore, the *Lyz2*$^{Cre}$; *Cflar*$^{f/f}$ model is also defined by an increase

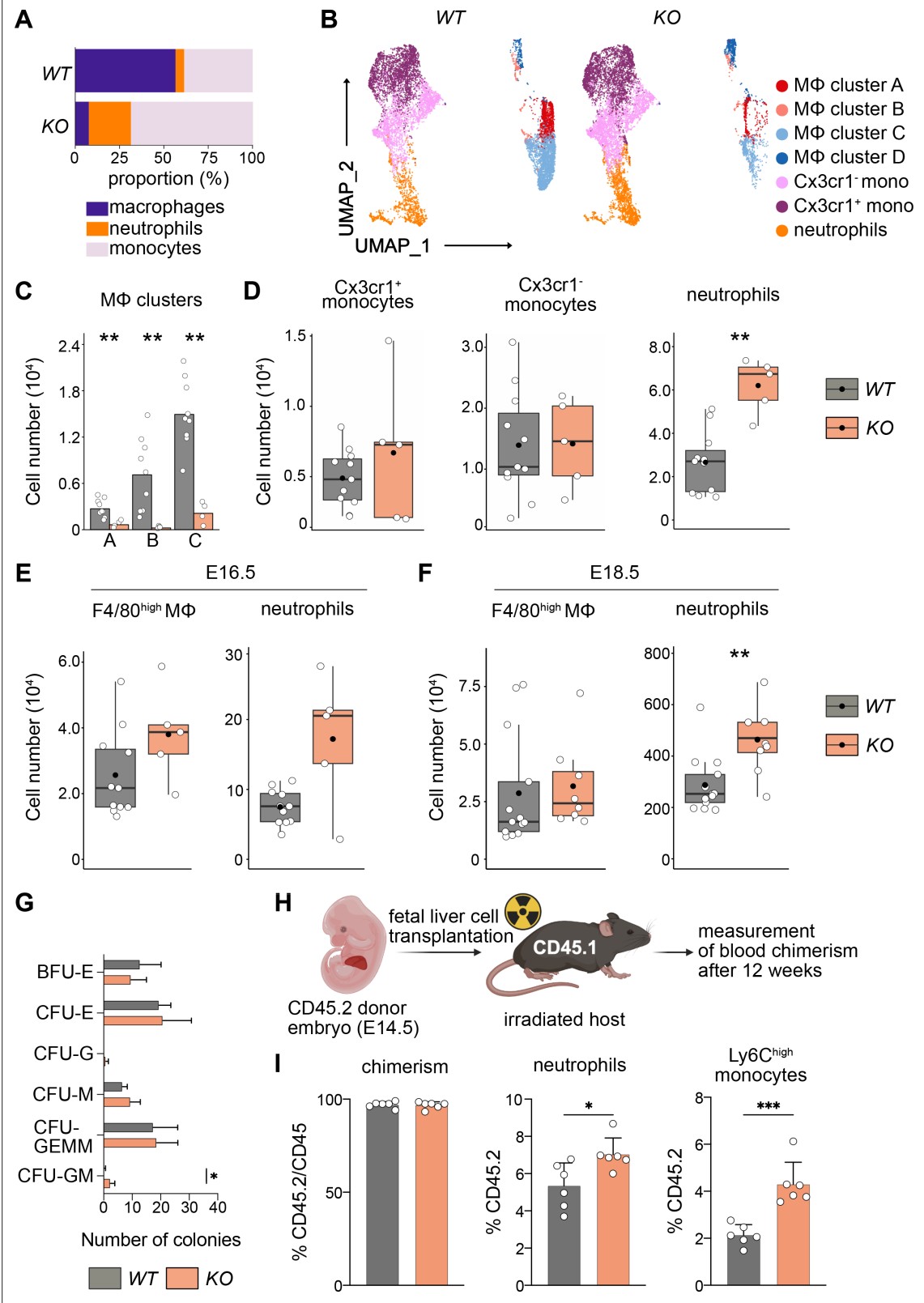

**Figure 7.** Macrophage depletion shifts hematopoiesis toward the granulocyte–macrophage progenitor (GMP) lineage. (**A**) Relative proportions of myeloid cells cells isolated from *Tnfrsf11a*[+/+]; *Spi1*[f/f] (*WT*) and *Tnfrsf11a*[Cre/+]; *Spi1*[f/f] (*KO*) fetal livers at E14.5. Average of *n* = 11 for *WT* and *n* = 5 for *KO*. (**B**) Flow-cytometry analysis of CD11b[low/+] cells isolated from *WT* and *KO* fetal livers at E14.5. Cell surface marker expression was used to generate unbiased clusters using UMAP, which were subsequently used for a gating strategy to quantify respective macrophage (M**Φ**), monocyte, and neutrophil

*Figure 7 continued on next page*

*Figure 7 continued*

populations. (**C**) Quantification of Mφ subclusters from *WT* and *KO* fetal livers at E14.5. *n* = 10 for *WT* and *n* = 4 for *KO*. (**D**) Quantification of monocytes (Cx3cr1⁺ and Cx3cr1⁻) and neutrophils from *WT* and *KO* fetal livers at E14.5. *n* = 11 for *WT* and *n* = 5 for *KO*. Quantification of F4/80 Mφ and neutrophils from *WT* and *KO* fetal livers at E16.5, *n* = 11 for *WT* and *n* = 5 for *KO* (**E**) and at E18.5, *n* = 13 for *WT* and *n* = 8 for *KO* (**F**). (**G**) Colony-forming unit assay from *WT* and *KO* fetal livers at E14.5. *n* = 4 for *WT* and *n* = 5 for *KO*. BFU: burst-forming erythroid, E: erythroid; G: granulocyte; GEMM; granulocyte, erythroid, macrophage, megakaryocyte; GM granulocyte, macrophage; M: macrophage. (**H**) Scheme of transplantation experimental setup. (**I**) Quantification of CD45.2 chimerism and % of neutrophils and Ly6C$^{high}$ monocytes in the blood of hosts 12 weeks after *WT* or *KO* fetal liver cells were transplanted. *n* = 6 for *WT* and *KO*. Wilcoxon test was performed comparing *WT* and *KO* cell number in (**C**), two-way analysis of variance (ANOVA) test was performed for the CFU assay in (**G**), unpaired Student's *t*-test was performed for adoptive transfer experiment in (I). \*\*\*p < 0.001, \*\*p < 0.01, and \*p < 0.05.

The online version of this article includes the following source data and figure supplement(s) for figure 7:

**Source data 1.** Quantified cell numbers of knockout and wildtype livers and cells from colony-forming unit (CFU) and adoptive transfer assays for *Figure 7*.

**Figure supplement 1.** Gating strategies for flow-cytometry data.

of inflammatory monocytes in the blood and spleen, indicating an alteration of myeloid progenitors leading to increased numbers of neutrophils and monocytes, likely driven by increased levels of G-CSF (*Gordy et al., 2011*). In contrast, the maintenance and longevity of neutrophils during embryogenesis are less well understood. Yet, published work suggests a different life span of fetal and adult neutrophils, with the E14.5 fetal liver harboring only a few apoptotic cells (*Liu et al., 2010*) in comparison to the adult spleen analyzed in *Lyz2$^{Cre}$; Cflar$^{f/f}$* mice and littermate controls (*Gordy et al., 2011*). Indeed, circulating neutrophils at E16.5 can be fate-mapped to an E8.5 yolk sac progenitor (*Gomez Perdiguero et al., 2015*), and during embryogenesis, there is rather a massive increase of neutrophils between E14.5 and E16.5 (*Freyer et al., 2021*) (and own data, not shown) instead of the steady-state turnover observed in adult mice. Thus, increased numbers of neutrophils in the fetal liver of *Tnfrsf11a$^{Cre/+}$; Spi1$^{f/f}$* embryos are unlikely caused by an increase of apoptotic neutrophils not being phagocytosed by macrophages.

Our RNA-seq data instead indicate that there is a transcriptional change in LT-HSCs when macrophages are lacking, supporting the hypothesis that fetal liver macrophages provide not only a niche for erythroblasts but also for LT-HSCs. In LT-HSC from *Tnfrsf11a$^{Cre/+}$; Spi1$^{f/f}$* fetal livers *Gata2* and *Gata3* were upregulated compared to littermate controls. Previous studies have shown the importance of *Gata2* and *Gata3* transcription factors in hematopoiesis (*Alsayegh et al., 2019*). GATA2 serves as a regulator of genes controlling the proliferative capacity of early hematopoietic cells during embryogenesis (*Tsai et al., 1994*; *Tsai and Orkin, 1997*) and the GMP cell fate (*Rodrigues et al., 2008*). Here, gene dosage is also crucial for HSC functionality since *Gata2* heterozygote (*Gata2$^{+/−}$*) mice displayed reduced GMP numbers in the bone marrow and serial replating CFU assays of *Gata2$^{+/−}$* bone marrow produced less GMPs compared to controls +/−. Furthermore, a study in GATA2-deficient human embryonic stem cells could show that GATA2 is required for the production of granulocytes (*Huang et al., 2015*). These data highlight the conserved function of *Gata2* in regulating HSC functionality on different levels, including their differentiation into granulocytes. Similar to Gata2, Gata3 can also regulate HSC maintenance and differentiation. Different studies of fetal and adult HSCs demonstrated that Gata3 is required for maintaining the self-renewing capacity of HSCs (*Fitch et al., 2012*; *Frelin et al., 2013*; *Ku et al., 2012*) and that expression of Gata3 is tightly regulating LT-HSCs to either self-renew or differentiate (*Frelin et al., 2013*). We could not observe an impact of *Gata2/Gata3* upregulation on LT-HSC numbers or their proliferation/serial replating capacity, suggesting a different mechanism in fetal livers. However, dysregulation of *Gata2* and/or *Gata3* expression may cause increasing numbers of GMPs and significantly more neutrophils in the fetal liver observed in *Tnfrsf11a$^{Cre/+}$; Spi1$^{f/f}$* embryos. Whether the proliferation and differentiation trajectories of fetal liver HSPCs are regulated by similar transcriptional networks as has been published for adult bone marrow remains to be investigated. Furthermore, using CD150 as an LT-HSC marker in the fetal liver may not exclude other MPPs, which have not been as systematically analyzed for the expression of surface markers and differential potential as the MPPs in the adult bone marrow (*Pietras et al., 2015*; *Sommerkamp et al., 2021*).Thus, to define the fetal LT-HSC niche in more detail, it will be important to examine whether macrophage-derived signals can control the expression and/or activity of Gata2 and Gata3 or other transcription factors that are known to control HSCs stem-cell ness and differentiation.

A study in zebrafish supports the notion of macrophage–HSC crosstalk requirement during development, uncovering a 'grooming' mechanism of embryonic macrophages that had a long-lasting impact on adult stem cells: HSPCs in the CHT often completed a cell division shortly after macrophage interactions and lack of CHT macrophages led to a decreased HSPC clonality in the adult marrow (*Wattrus et al., 2022*). Interestingly, their data suggest that HSPC proliferation in the CHT is mediated through extracellular signal-regulated kinase/mitogen-activated protein kinase (ERK/MAPK) activity, which is controlled by macrophage-derived Il1b (*Wattrus et al., 2022*). Indeed, the scRNA-seq macrophage cluster 2 is specifically expressing Il1b and could have a similar effect on a subset of LT-HSCs in the mouse fetal liver. Therefore, dissecting the macrophage-derived ligands and their effect on HSPC populations on a single-cell level will shed light on the crosstalk mechanisms in mouse fetal livers in the future.

Defining factors that control the HSC niche is essential to support efforts for an in vitro expansion and targeted differentiation of HSCs. Only a few studies have focused on macrophages as HSC niche cells so far, as they are primarily viewed as interacting cells of erythroblasts in both the fetal liver and the adult bone marrow. Here, we show that macrophages provide a niche for LT-HSCs in the fetal liver, and that macrophage deficiency leads to changes in the LT-HSC transcriptional program and their differentiation capacity. Likely, not only the immediate interaction but also macrophage-derived cytokines and growth factors affect processes, such as fate decisions and stem-cell ness. Our results provide a starting point to study the impact of macrophage-derived signals on LT-HSC functionality during embryogenesis and adulthood.

# Materials and methods

**Key resources table**

| Reagent type (species) or resource | Designation | Source or reference | Identifiers | Additional information |
|---|---|---|---|---|
| Commercial assay or kit | CODEX conjugation kit | Akoya Biosciences | 7000008 | |
| Commercial assay or kit | CODEX staining kit | Akoya Biosciences | 7000009 | |
| Commercial assay or kit | CODEX Buffer | Akoya Biosciences | 7000001 | |
| Chemical compound, drug | BS3 fixative | Thermo Fisher Scientific | 21580 | |
| Chemical compound, drug | CollagenaseD | Sigma-Aldrich | 11088882001 | |
| Chemical compound, drug | DAPI (4',6-diamidino-2-phenylindole, dilactate) | BioLegend | 422801 | 1:10,000 |
| Chemical compound, drug | DNase | Sigma-Aldrich | DN25-1G | |
| Chemical compound, drug | DRAQ7 | BioLegend | 424001 | 1:1000 |
| Chemical compound, drug | Fetal calf serum | Bio&Sell | FBS. S 0615HI | |
| Chemical compound, drug | Giemsa solution | Merck | 109204 | |
| Chemical compound, drug | May–Grünwald solution | Merck | 101424 | |
| Chemical compound, drug | MethoCult | StemCell Technologies | 3434 | |
| Chemical compound, drug | Normal Goat Serum | VWR | ICNA 08642921 | |
| Chemical compound, drug | Paraformaldehyde (PFA) | Thermo Fisher Scientific | 11586711 | |
| Chemical compound, drug | Progesterone | Sigma-Aldrich | P3972-5G | |
| Chemical compound, drug | QIAzol Lysis Reagent | QIAGEN | 79306 | |
| Chemical compound, drug | Rat serum | Bio-Rad | C13SD | |
| Chemical compound, drug | Roswell Park Memorial Institute medium (Seahorse XF RPMI medium) | Agilent Technologies | 103576-100 | |
| Chemical compound, drug | Tamoxifen | Sigma-Aldrich | T5648 | |
| Chemical compound, drug | Weise buffer tablet | Merck | 109468 | |

*Continued on next page*

*Continued*

| Reagent type (species) or resource | Designation | Source or reference | Identifiers | Additional information |
|---|---|---|---|---|
| Software | CODEX instrument manager | Akoya Biosciences | | |
| Software | CODEX Processor | Akoya Biosciences | | |
| Software | CODEX MAV | Akoya Biosciences | | |
| Software | QuPath | *Bankhead et al., 2017* | | |
| Software | ImageJ | *Schindelin et al., 2012* | | |
| Software | FlowJo | BD | v.10.8.1 | |

### Experimental mice

All mice were maintained on a C57BL/6 background and housed in SPF conditions. Animal procedures were performed in adherence to our project license 2018.A056 issued by the 'Landesamt für Natur, Umwelt und Verbraucherschutz' (LANUV). Whenever possible, The ARRIVE guidelines 2.0 were followed (The ARRIVE Essential 10, https://arriveguidelines.org/arrive-guidelines). *Tnfrsf11a$^{Cre/+}$; Spi1$^{f/+}$* males were crossed to *Spi1$^{f/f}$* females to generate embryos lacking macrophages. *Tnfrsf11a$^{Cre/+}$, Ms4a3$^{Cre/+}$*, and *Cxcr4$^{CreERT/+}$* mice were crossed to the *Rosa26$^{YFP}$* strain to generate embryos suitable for the fate-mapping. Adult mice were mated overnight to obtain embryos. To fate-map HSCs, *Cxcr4$^{CreERT}$; Rosa26$^{YFP}$* embryos were pulsed using 4-hydroxytamoxifen injection (75 mg/kg) at embryonic day (E)10.5. To prevent tamoxifen-related abortions, progesterone (37.5 mg/kg) was injected simultaneously with tamoxifen into the mice. The female was examined for vaginal plug formation the next day and the embryos were considered to be E0.5.

### In vivo engraftment of fetal liver cells

Donor mice were with *Ly5.2* (CD45.2) congenic background. *Spi1$^{f/f}$* females were mated overnight with *Tnfrsf11a$^{Cre/+}$; Spi1$^{f/+}$* males and embryonic development was estimated considering the day of vaginal plug formation as 0.5 days post-coitum (E0.5). E14.5 embryos were harvested and kept on ice in PBS while being genotyped for *Spi1* and *Tnfrsf11a* loci. Fetal livers from *Spi1$^{f/f}$* and *Tnfrsf11a$^{Cre/+}$; Spi1$^{f/f}$* were then isolated, homogenized, incubated at 37°C for 20 min in 300 µl digestion mix (1.5% fetal calf serum (FCS), 1 mg/ml collagenase D, and 100 U/ml DNase I), filtered through a 100-µm strainer and resuspended in PBS to achieve a single-cell suspension at a concentration of $10^7$ cells/ml. $10^6$ total fetal liver cells were injected into lethally irradiated (10 Gray) *Ly5.1* (CD45.1) mice through the orbital vein 8 hr after irradiation. Engraftment was quantified 12 weeks after transplantation by flow cytometric analysis of the blood.

### Cryosection and whole-mount immunostaining

Pregnant mice were sacrificed through cervical dislocation. Embryos at E14.5 were harvested and stored in cold 1× Dulbecco's phosphate-buffered saline (DPBS w/o Ca and Mg, PAN-Biotech) and dissected under a Leica M80 microscope. Fetal livers were harvested and fixed in 1% paraformaldehyde overnight at 4°C for immunofluorescence staining. Fixed lobes were washed three times for 10 min with DPBS, incubated in 30% sucrose. The fixed lobes were dehydrated using increasing methanol (Fisher Scientific) gradient diluted in DPBS. Samples were incubated with primary antibodies (AB) (*Supplementary file 5*) overnight at 4°C in 0.4% PBT (DPBS with 0.4% Triton X-100). Afterward, the samples were washed three times for 10 min in washing buffer DPBS with 0.2% Triton X-100, 3% NaCl (Grüssing) at room temperature. The same procedure was repeated using the secondary antibodies (*Supplementary file 5*) and, if needed, using the directly conjugated antibodies a third time. samples were cleared in benzyl-alcohol benzyl-benzoate (BABB 1:2 proportion). Fetal liver lobes were placed for 30 min in 50% BABB, followed by incubation in 100% BABB for 30 min to obtain transparent tissues. The samples were placed in a cavity slide (Brand) filled with BABB. A round cover glass was carefully placed on the tissue and sealed using nail polish. The samples were 3D visualized using LSM 880 Zeiss confocal microscope with a ×63 (oil) objective.

### Volumetric imaging and analysis

Fetal livers were harvested from E14.5 embryos as described above and fixed overnight at 4°C in Fix/Perm buffer at a dilution of 1:4 (BD Bioscience). Liver lobules were incubated for 20 min at 37°C

in HistoReveal (Abcam) diluted 1:1 with phosphate-buffered saline (PBS) and then washed in PBS. Samples were incubated overnight in a permeabilization solution (PBS, 0.1% Triton-X, 0.1 M glycine, 5% bovine serum albumin (BSA), 5% goat serum) at room temperature and then stained overnight in staining buffer (PBS, 0.1% Triton-X, 1% BSA) at room temperature as follows: DAPI (1:10,000 dilution; Thermo Fisher), Alexa Fluor 555 rabbit anti-mouse Iba1 (1:200 dilution; Cell Signaling, #36618), CD150 unconjugated (1:100 dilution; Invitrogen, clone 9D1). After washing three times in wash buffer (PBS, 0.1% Triton-X), a further overnight incubation was performed in staining buffer at 4°C with the following staining: Alexa Fluor 647 fab fragment goat anti-rat IgG (Jackson ImmunoResearch, #112-607-003). After further washing, the tissue was optically cleared in Ce3D (BioLegend; #427702) for 1 hr and mounted for immediate microscopy. Deep multiplex imaging of optically cleared liver tissues was performed with a Zeiss LSM 880 NLO confocal microscope and a 'Plan-Apochromat' objective (×20) with an NA of 1.0. Imaging was performed in tiles at a zoom of 3.0 in stacks of 150–200 μm with a step size of 0.5 μm. After stitching (Imaris Stitcher, Bitplane), a pixel classifier (Ilastik) was trained for semantic segmentation of nucleated, Iba1$^+$ or CD150$^+$ cells, which was then used for cellular segmentation using the IMARIS volume segmentation pipeline (IMARIS 10.0, Bitplane). Analysis of the spatial relationships of Iba1$^+$ and CD150$^+$ was performed using the IMARIS module Vantage, to automatically extract the distances between the cells and the mutual attraction and repulsion effects (*Gomariz et al., 2018*).

## Co-detection by indexing

5 μm slices of fetal liver from E14.5 wildtype embryos were prepared and used for CODEX staining following the manufacturer's instructions. Briefly, sections were retrieved from the freezer, let dry on drierite beads, and fixed for 10 min in ice-cold acetone (Sigma-Aldrich, St. Louis, MO, USA). After fixation, samples were rehydrated and photobleached twice as described in *Du et al., 2019*. Following photobleaching, sections were blocked and stained with a 20-plex CODEX antibody panel (*Supplementary file 5* and *Supplementary file 6*) overnight at 4°C. After staining, samples were washed, fixed with ice-cold methanol, washed with 1× PBS, and fixed for 20 min with BS3 fixative (Sigma-Aldrich, St. Louis, MO, USA). A final washing step with 1× PBS was performed.

A multicycle CODEX experiment was performed following the manufacturer's instructions. Images were acquired with a Zeiss Axio Observer widefield fluorescence microscope using a ×20 objective (NA 0.85) and z-spacing of 1.5 μm. The 405, 488, 568, and 647 nm channels were used. After imaging, raw files were exported using the CODEX Instrument Manager (Akoya Biosciences, Marlborough, MA, USA) and processed with CODEX Processor v1.7 (Akoya Biosciences). Image processing included background subtraction using the DAPI signals of the first and last empty cycles of the acquisition, deconvolution, shading correction, and stitching. For cell segmentation, DAPI counterstain was used for object detection, whereas sodium-potassium ATPase antibody staining was used as a membrane marker for delineating the cell shape.

A manual cell classification was performed in CODEX MAV 1.5 (Akoya Biosciences). Annotation of the macrophage clusters was done using the same gating strategy as in flow cytometry, with the difference that F4/80$^+$CD11b$^+$ cells were not gated but F4/80$^+$ Iba1$^+$ cells. HSCs were gated as CD150$^+$ c-Kit$^+$ cells, erythrocytes as CD45$^-$Ter119$^+$ cells, and blood vessels as CD45$^-$CD31$^+$SMA$^+$ cells. After cell classification, Voronoi diagrams were generated in CODEX MAV using the four macrophage clusters, blood vessels, and HSCs as seeds.

## Spatial analyses and determination of cellular neighborhoods with CODEX images

Log Odd Ratio analysis for spatial interactions was performed in CODEX MAV. For this, after cell classification, the four macrophage populations, HSCs, and blood vessels were selected. The selected minimum and maximum distances of interaction were 5 and 50 μm, respectively.

For cellular neighborhood analyses, the .csv files generated with CODEX MAV were exported to CytoMAP (*Stoltzfus et al., 2020*) and the same cell classification was used to annotate the cells. A raster scan with a radius of 50 μm was performed to spatially segment the image. To define the cellular neighborhoods based on local composition, an SOM clustering algorithm was used, considering only the macrophage populations. Heatmaps were generated to determine the cell composition of each neighborhood. To measure the distances between macrophage clusters and erythrocytes, images

were exported to QuPath v0.3 and cells were detected using DAPI signals. Single object classifiers for each marker were trained, and these were used to generate composite classifiers to identify macrophage populations, erythrocytes, and HSCs as before. The distance between the cells of each macrophage cluster and their closest erythrocyte was measured and plotted.

To validate the proximity of macrophage clusters to HSCs, images were exported to QuPath v0.3, cells were segmented, and HSCs were identified, as described above. A circle with a fixed radius of 50 μm was drawn and centered on 20 randomly selected HSCs. Next, the same composite classifiers to identify macrophage clusters were applied to the annotated circles, and the number of cells of each macrophage cluster within the defined radius was counted.

To determine the percentage of CD41$^+$/CD150$^+$ cells, three independent sections were analyzed using CODEX MAV 1.5. First, CD150$^+$ objects were manually gated. Subsequently, from this gate, CD41$^+$ were gated as well, to quantify double positive objects. Single- and double-positive objects were plotted.

To quantify the colocalization of paired macrophage markers, the same representative area of a section was selected and the raw images of each marker were uploaded to ImageJ and analyzed using the JACoP plugin, as described before (*Bolte and Cordelières, 2006*).

## CFU assay

CFU assays were performed according to the Mouse 'Colony-Forming Unit' (CFU) Assays Using 'Metho-Cult' GF M3434 protocol (StemCell Technologies) (STEMCELL). Briefly, a fetal liver lobe was collected in fluorescence cell sorting (FACS) buffer (1× DPBS with 2% 100 mM ethylenediaminetetraacetic acid (Sigma-Aldrich), 0.5% BSA) and digested using digestion solution (1% DNase (Sigma-Aldrich), 1% collagenase D (Sigma-Aldrich), 3% FCS (Bio&Sell), 1× DPBS). The samples were mechanically disrupted and incubated for 30 min at 37°C. Following the digestion, the whole volume was transferred in an FACS tube and centrifuged for 5 min at 400 × $g$, 4°C. The supernatant was discarded, and the pellet was resuspended in 1 ml sterile Roswell Park Memorial Institute medium (RPMI, supplemented with 10% FCS, 1% penicillin–streptomycin, 1% D-glutamate, 1% pyruvate). 3 × $^5$ live cells were taken from the suspension and filled up to 1 ml with RPMI to achieve a 3 × 10$^5$ cells/ml concentration. 1.5 ml MethoCult aliquots (StemCell Technologies) were thawed at room temperature and vortexed vigorously. The cell suspension was added to the MethoCult in a way to achieve 3 × 10$^4$ cells/ml concentration. 1 ml MethoCult mixed with the cell suspension was slowly taken up using a pipette and then transferred to a 35-mm cell culture dish (VWR). The cell culture dish was cautiously tilted until it was covered with medium and was put inside the incubator. After 12 days of cell culture, colonies were first identified by their phenotype, that is their size, shape, and density were analyzed based on representative pictures shown in the CFU assay protocol by STEMCELL Technologies. Subsequently, colonies were picked and identification was validated by the May–Grünwald–Giemsa staining (Merck) of the colonies. Prior to the staining, the colonies were picked up using a 10-μl pipet under Leica M80 microscope and collected into microtubes containing 10 μl FACS buffer. The collected colonies were transferred on slides using cytospin funnels (Hettich), centrifuged at 800 RPM by a cytospin centrifuge (Hettich) for 10 min. After the centrifuge the slides were air-dried and fixed using cold methanol.

## Flow-cytometry sample preparation and data acquisition

Pregnant mice were sacrificed through cervical dislocation at E14.5. The fetal liver, brain, and lung were harvested from the embryos of the *Tnfrsf11a$^{Cre}$; Spi1$^{flox/+}$*. The collected tissues were digested for 20 min at 37°C in a digestion solution. The cell suspensions were centrifuged and the supernatants were removed. The pellets were resuspended in 50 μl blocking solution (2% rat serum) for 10 min incubation on ice. The volume of each sample of the *Tnfrsf11a$^{Cre}$; Spi1$^{flox/+}$* model was measured to obtain the total cell number. The samples were stained with primary antibodies for 25 min (*Supplementary file 5*). Afterward, the samples were washed by adding 100 μl FACS to the suspension and centrifuge at 400 × $g$ for 5 min at 4°C. The same procedure was repeated for the secondary antibodies. Finally, the cells were stained with Hoechst live/dead staining (1:10,000) before flow-cytometry and recorded using a FACSymphony (BD Biosciences) cytometer.

The same procedure was also done with harvested fetal livers from wildtype embryos at E14.5. These samples were stained with primary and secondary antibodies designated to investigate the

heterogeneity of macrophages (*Supplementary file 5*). The cells were stained with DRAQ7 live/dead staining (1:1000) before flow-cytometry and recorded using a FACSymphony cytometer.

## Analysis of flow-cytometry data for quantification of cells

Flow-cytometry data analysis was performed using FlowJo Software v.10.8.1 Becton, Dickinson, and Company. The count of each cell type was recorded and plotted using R (v. 4.0.5) and the ggplot2 (v. 3.3.5) and ggpubr (v. 0.4.0) (*Wickham, 2016*, *Kassambara, 2020*).

## Analysis of flow-cytometry data for heterogeneity of macrophages

The CD11b$^+$ F4/80$^+$ cells were gated (*Figure 1—figure supplement 1A*) and downsampled using downsample plug-in (v.3.3.1) in Flowjo. The downsampled population was imported and analyzed in R (https://www.r-project.org/ v. 4.0.5). The importing and processing of data were done using the CATA-LYST package (v. 1.18.1) (*Crowell et al., 2020*), which was installed through the Bioconductor package (v 3.14). The visualization of data was done using the UMAP algorithm (*McInnes et al., 2018*), and the clustering of data was done using FlowSOM (*Van Gassen et al., 2015*) clustering and ConsensusClusterPlus metaclustering (*Wilkerson and Hayes, 2010*). The resulting clusters were manually inspected for expression of different markers and clusters of interests were subset and merged if necessary to form final clusters that would represent the macrophages and their heterogeneity.

## Cell sorting of HSCs and macrophages for RNA-sequencing

Pregnant mice were sacrificed through cervical dislocation at embryonic day E14.5. The fetal liver was collected from *Tnfrsf11a$^{Cre/+}$; Spi1$^{f/f}$* and wildtype mouse models' embryos. The collected tissues were digested for 20 min at 37°C in a digestion solution. The cell suspensions were centrifuged and the supernatants were removed. The pellets were resuspended in 50 µl blocking solution (2% rat serum) for 10 min incubation on ice. The samples were washed and stained for 25 min. At the end of incubation, the samples were washed and centrifuged at 400 × *g* for 5 min at 4°C. Finally, the pellet was resuspended in FACS buffer. The cells were stained with DAPI live/dead staining in a final dilution 1:10,000 before flow-cytometry analysis using BD FACS ARIA III. ~700–1200 LT-HSCs were sorted according to the gating strategy (*Figure 6—figure supplement 1*) into microtubes containing 500 µl of Qiazol lysis buffer while the CD11b$^{low/+}$ F4/80$^+$ cells (*Figure 1—figure supplement 1*) were sorted into microtubes containing 100 µl of FACS buffer. The cells were used for loading the arrays in the next steps for single-cell RNA-sequencing.

## Bulk RNA-sequencing and analysis

Ten samples in total (four controls and six knockouts) were analyzed for the bulk RNA-sequencing. Total RNA was extracted using the miRNeasy micro kit (QIAGEN) and quantified via RNA assay on a tape station 4200 system (Agilent). 5 ng total RNA was used as an input for library generation via SmartSeq 2 (SS2) RNA library production protocol as previously described (*Picelli et al., 2014*). Pre-amplification polymerase chain reaction (PCR) was performed with 16 cycles for samples. Libraries were quantified using the Qubit HS dsDNA assay (Invitrogen), and fragment size distribution was analyzed via D1000 assay on a tape station 4200 system (Agilent). SS2 libraries were sequenced single-end with 75 cycles on a NextSeq2000 system using P3 chemistry (Illumina). Samples were demultiplexed and fastq files were generated using bcl2fastq2 v2.20 before alignment and quantification using Kallisto v0.44.0 based on the Gencode (mm10, GRCm38) vM16 (Ensembl 91) reference genomes. Sequencing results from the experiments were pseudoaligned using the Kallisto tool set. The counts were imported into R and analyzed using the DEseq2 package (*Love et al., 2014*). The genes with less than 11 counts in all samples were removed. The counts were transformed using the variance Stabilizing Transformation (VST) function of the DEseq2 pipeline. The knockout samples were compared to the control samples during the analysis, and genes were ranked on the differential expression (log fold-change (LFC) threshold of 0.1, adjusted p-value Benjamini-Hochberg (BH) <0.1). The ranked gene list was divided into down- and upregulated genes (Logfold2change >0.5 and <−0.5) and used for GO analysis using the clusterProfiler package (*Yu et al., 2012*). The DEGs were visualized using volcano plot using Enhancedvolcano package (*Blighe et al., 2019*).

## Single-cell RNA-sequencing

Seq-Well arrays were prepared as described by *Gierahn et al., 2017*. Seq-Well libraries were generated as described by *Gierahn et al., 2017*, *Hughes et al., 2020*. The cDNA libraries (1 ng) were tagmented

with home-made single-loaded Tn5 transposase in TAPS–DMF buffer (50 mM TAPS–NaOH (pH 8.5), 25 mM $MgCl_2$, 50% DMF in $H_2O$) for 10 min at 55°C and the tagmented products were cleaned with the MinElute PCR kit following the manufacturer's instructions. Finally, a master mix was prepared (2X NEBNext High Fidelity PCR Master Mix, 10 µM barcoded index primer, 10 µM P5-SMART-PCR primer) and added to the samples to attach the Illumina indices to the tagmented products in a PCR reaction (72°C for 5 min, 98°C for 30 s, 15 cycles of 98°C for 10 s, 63°C for 30 s, 72°C for 1 min). The pools were cleaned with 0.6× volumetric ratio AMPure XP beads. The final library quality was assessed using a High Sensitivity DNA5000 assay on a Tapestation 4200 (Agilent) and quantified using the Qubit high-sensitivity dsDNA assay. Seq-Well libraries were equimolarly pooled and clustered at 1.4 pM concentration with 10% PhiX using High Output v2.5 chemistry on a NextSeq500 system. Sequencing was performed paired-end using custom Drop-Seq Read 1 primer for 21 cycles, 8 cycles for the i7 index, and 61 cycles for Read 2. Single-cell data were demultiplexed using bcl2fastq2 (v2.20). Fastq files from Seq-Well were loaded into a snakemake-based data pre-processing pipeline (version 0.31, available at https://github.com/Hoohm/dropSeqPipe, copy archived at *Roelli, 2023*) that relies on the Drop-seq tools provided by the McCarroll lab (*Macosko et al., 2015*). STAR alignment within the pipeline was performed using the murine GENCODE reference genome and transcriptome mm10 release vM16.

## Analysis of single-cell RNA-sequencing

The single-cell RNA was analyzed using the scanpy package (v.1.8.1) (*Wolf et al., 2018*) in python (v.3.4.1) (*Van Rossum and Drake, 1995*). The cells were pre-processed and filtered by checking for cells expressing less than 200 genes in less than three cells. After the filtration, cells were processed further and clustered using the Leiden algorithm (*Traag et al., 2019*). The clusters were investigated using their DEGs. The DEG lists were identified using Wilcoxon-Signed-Rank Test by comparing each cluster to the rest of the clusters. The selection of the clusters of interest was done in an iterative way. Three clustering steps were performed in total to subset the cells. Briefly, the first round of clustering was done in a none-stringent manner and the resulted groups were monitored for their DEGs and were assigned to certain cell types that they resembled most. Based on the DEGs the clusters 1, 4, 5, and 9 were containing macrophages while the other groups were representing other cell types and states. This selection was assured further by exploring the expression of a set of pMac markers among the selected clusters. Clusters that could contain macrophages were subset for a second round of clustering. The subset cells were processed and clustered for a second time to have more homogenous cells regarding cell types. Again, the clusters were subjected to another round of selection using two different sets of signature markers for genes that are expressed in macrophages. The selected clusters in the second round were subset and the cells were processed for a final round of clustering resulting in distinguishable clusters of macrophage cells and their precursors from the rest. The finals clusters were analyzed and the DEGs for the were extracted A psuedotime analysis using the PAGA method (*Wolf et al., 2019*) was later performed to analyze the trajectory of these groups.

## Correlation matrix

To make a correlation between the single-cell RNA-seq data and the flow-cytometry data, the clusters of interest from both of the datasets were extracted. The expression values of mutual markers between them were scaled and normalized for each of the two datasets. The correlation between the two datasets was calculated using spearman's rank correlation coefficient and results were visualized using the pheatmap package (*Kolde and Kolde, 2015*) in R.

## Ligand–receptor analysis

CellTalk database information (*Shao et al., 2021*) was download and the ligands list were explored among the expressed genes of the five final macrophage clusters and the ligands that were expressed were selected (208 ligands). A GO analysis using ClusterProfiler package was done on the selected ligand's genes. Terms that were associated with the hematopoiesis were selected and the genes belonging to these terms were extracted. Genes with a minimum of five counts (single-cell data) among the three final macrophage clusters were chosen (100 ligands). Their corresponding receptors were taken from the CellTalk database. The expression of those receptors was explored in the bulk RNA-seq data from control LT-HSCs and the receptors that had more than average of 40 counts

were selected. The ligand and receptor interactions were visualized using a circular plot using circlize package (*Gu et al., 2014*).

## Acknowledgements

We thank Cornelia Cygon for technical support and Florent Ginhoux for providing Ms4a3^Cre mice. The work was funded by the Deutsche Forschungsgemeinschaft (DFG, German Research Foundation) under Germany's Excellence Strategy-EXC2151-390873048 (to EM, JLS, EK, MB, and AS), GRK2168 (to EM and KM), GRK1873/2 (to EM and NM), SFB 1454 - Project ID 432325352 – (to EM, JLS, AS, and MB), FOR5547 – Project-ID 503306912 (to EM), by Boehringer Ingelheim Fonds (doctoral fellowship to KM). EM is supported by the European Research Council (ERC) under the European Union's Horizon 2020 research and innovation program (Grant Agreement No. 851257). SU is supported by DFG (project-IDs 448121430, 405969122, 505539112), the Hightech Agenda Bavaria and by an ERC starting grant (project-ID 101039438). Volumetric liver imaging was performed on a DFG-funded confocal microscope (project-ID 261193037). This work was supported by the Open Access Publication Fund of the University of Bonn.

## Additional information

### Funding

| Funder | Grant reference number | Author |
|---|---|---|
| Deutsche Forschungsgemeinschaft | EXC2151-390873048 | Joachim L Schultze<br>Eva Kiermaier<br>Marc Beyer<br>Andreas Schlitzer<br>Elvira Mass |
| Deutsche Forschungsgemeinschaft | GRK2168 | Katharina Mauel<br>Elvira Mass |
| Deutsche Forschungsgemeinschaft | GRK1873/2 | Elvira Mass |
| Deutsche Forschungsgemeinschaft | SFB1454 | Joachim L Schultze<br>Marc Beyer<br>Andreas Schlitzer<br>Elvira Mass |
| Deutsche Forschungsgemeinschaft | FOR5547 - Project-ID 503306912 | Elvira Mass |
| Boehringer Ingelheim Stiftung | | Katharina Mauel |
| Horizon 2020 Framework Programme | 851257 | Elvira Mass |
| Deutsche Forschungsgemeinschaft | 448121430 | Stefan Uderhardt |
| Deutsche Forschungsgemeinschaft | 405969122 | Stefan Uderhardt |
| Deutsche Forschungsgemeinschaft | 505539112 | Stefan Uderhardt |
| Hightech Agenda Bavaria | | Stefan Uderhardt |
| Horizon 2020 Framework Programme | 101039438 | Stefan Uderhardt |
| Deutsche Forschungsgemeinschaft | 261193037 | Stefan Uderhardt |

The funders had no role in study design, data collection, and interpretation, or the decision to submit the work for publication.

## Author contributions
Amir Hossein Kayvanjoo, Data curation, Formal analysis, Supervision, Investigation, Visualization, Writing - original draft, Writing - review and editing; Iva Splichalova, Hao Huang, Katharina Mauel, Data curation, Formal analysis, Investigation, Visualization, Writing - review and editing; David Alejandro Bejarano, Data curation, Formal analysis, Investigation, Visualization, Writing - original draft, Writing - review and editing; Nikola Makdissi, David Heider, Hui Ming Tew, Data curation, Formal analysis, Writing - review and editing; Nora Reka Balzer, Investigation, Writing - review and editing; Eric Greto, Data curation, Formal analysis, Investigation; Collins Osei-Sarpong, Kevin Baßler, Data curation, Methodology, Writing - review and editing; Joachim L Schultze, Resources, Supervision, Writing - review and editing; Stefan Uderhardt, Data curation, Formal analysis, Supervision, Investigation, Methodology, Writing - original draft, Writing - review and editing; Eva Kiermaier, Writing - review and editing; Marc Beyer, Resources, Supervision, Methodology, Writing - review and editing; Andreas Schlitzer, Resources, Data curation, Supervision, Methodology, Writing - original draft, Writing - review and editing; Elvira Mass, Conceptualization, Resources, Data curation, Formal analysis, Supervision, Funding acquisition, Investigation, Visualization, Methodology, Writing - original draft, Project administration, Writing - review and editing

## Author ORCIDs
Amir Hossein Kayvanjoo (ID) http://orcid.org/0000-0003-1315-8336
Hao Huang (ID) http://orcid.org/0000-0003-3878-3947
Nikola Makdissi (ID) http://orcid.org/0000-0002-9345-038X
Nora Reka Balzer (ID) http://orcid.org/0000-0001-9895-7051
Kevin Baßler (ID) http://orcid.org/0000-0002-4780-372X
Eva Kiermaier (ID) http://orcid.org/0000-0001-6165-5738
Elvira Mass (ID) http://orcid.org/0000-0003-2318-2356

## Ethics
This study was performed in strict accordance with the recommendations of the LANUV and Veterinary Office of the City of Bonn. All of the animals were handled according to approved institutional animal care at the LIMES GRC. The protocol was approved by the LANUV (Permit Number: 2018. A056).

## Decision letter and Author response
Decision letter https://doi.org/10.7554/eLife.86493.sa1
Author response https://doi.org/10.7554/eLife.86493.sa2

# Additional files

## Supplementary files
• Supplementary file 1. Macrophage subcluster-specific gene set enrichment analysis (GSEA).

• Supplementary file 2. Ligands expressed by macrophages at E14.5.

• Supplementary file 3. Differentially expressed genes in knockout versus wildtype long-term hematopoietic stem cells (LT-HSCs).

• Supplementary file 4. GO term analysis of knockout versus wildtype long-term hematopoietic stem cells (LT-HSCs).

• Supplementary file 5. Antibody list.

• Supplementary file 6. Barcodes and reporters for co-detection by indexing CODEX.

• MDAR checklist

## Data availability
RNA-seq data from bulk and single-cell experiments are available under GEO accession number GSE225444. Source data for CODEX pictures (raw .tiff files) and analyses are available as pyramidal file at Dryad (https://doi.org/10.5061/dryad.fn2z34v00). Due to size restrictions, the original CODEX .czi files could not be uploaded, but will be made available without restrictions after contacting the corresponding author.

The following datasets were generated:

| Author(s) | Year | Dataset title | Dataset URL | Database and Identifier |
|---|---|---|---|---|
| Mass E | 2024 | Source Data Kayvanjoo et al. CODEX of fetal livers at E14.5 | https://dx.doi.org/10.5061/dryad.fn2z34v00 | Dryad Digital Repository, 10.5061/dryad.fn2z34v00 |
| Kayvanjoo AM | 2023 | Fetal liver macrophages contribute to the hematopoietic stem cell niche by controlling granulopoiesi | https://www.ncbi.nlm.nih.gov/geo/query/acc.cgi?acc=GSE225444 | NCBI Gene Expression Omnibus, GSE225444 |

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
