## [Editor Report]

Using single-cell sequencing, high-resolution imaging, and inducible genetic deletion of yolk-sac (YS) derived macrophages, the authors present a useful map of fetal liver macrophage subpopulations and provide important data demonstrating that heterogeneous fetal liver macrophages regulate erythrocyte enucleation, interact physically with fetal HSCs, and may regulate neutrophil accumulation in the fetal liver. These important findings provide a solid foundation for further investigating the effects of macrophages on HSC function during fetal hematopoiesis and into adulthood and will be useful for the field of macrophage biology and developmental hematopoiesis.

---

## [Decision Letter]

**Decision letter after peer review:**

Thank you for submitting your article "Fetal liver macrophages contribute to the hematopoietic stem cell niche by controlling granulopoiesis" for consideration by *eLife*. Your article has been reviewed by 2 peer reviewers, and the evaluation has been overseen by a Reviewing Editor and Mone Zaidi as the Senior Editor. The reviewers have opted to remain anonymous.

Essential revisions (for the authors):

The reviewers concluded that although the strength of the study lies in the characterization and categorization of distinct FL macrophages, and in the fate-mapping aspect, the conclusion that macrophages are regulating the hematopoietic stem cell niche and granulopoiesis is not well-supported by the data as presented. Either transplantation or co-culture experiments at a minimum would be necessary to demonstrate an effect on hematopoiesis and lineage differentiation. Alternatively, strongly adapting the language/the conclusion on the influence of the macrophages on HSCs in the fetal liver might improve the study.

*Reviewer #1 (Recommendations for the authors):*

Based on initial clustering, it seems that many other macrophages within the fetal liver are excluded from further downstream analysis. It would have been very interesting and important to investigate both fate-mapping and gene expression signatures in other clusters besides those that specifically align with pMAC signatures.

The authors might have detected the effects of macrophage depletion on HSC function if they had isolated LT-HSCs and performed transplantation assays. This would have provided much stronger evidence supporting the author's conclusion of granulocytic bias. Colony-forming assays, as presented, are not convincing of the role of fetal macrophages in regulating HSC output or function, particularly in the absence of any effects in vivo in the conditional deletion model.

Also, it may have been too early for LT-HSCs to produce granulocytes, and for them to significantly accumulate in the FL. This could be directly tested using the fate-mapping models on hand.

In general, some of the data could be more clearly presented. For example, frequencies should be shown in 1i. Data in Figure 3E don't represent data in Figure 3F – e.g., there appear to be as many macrophages of each subtype in proximity to each HSC in 3E. In 3C, Cluster C is not visible. In Figure 6, differences in cluster size between KO and WT should be quantified. These minor changes would help clarify some of the data.

*Reviewer #2 (Recommendations for the authors):*

Figure 1 and Figure S1. The analysis and identification of macrophages in the fetal liver is strong. The initial selection step reduces the number of clusters further analyzed from 11 to 4, simply based on the pMac signature, plus a subsequent reduction (Figure S1D). How robust is that reduction, as strongly influences all downstream analyses and reduces the number of cells for subsequent analyses? What might then be cells not included?

Is the analysis robust enough to then include all types of macrophages, or the majority populations?

To this reviewer, the presentation of the alignment of the flow-based analysis (informed by the gene expression data) that was also reduced in complexity aligned that likely well to the gene expression clusters could be further improved and supported by additional information and steps much better put into perspective. This alignment is critical as protein and gene-based clustering seem to align, which is central due to the high level of reduction that was applied in obtaining the results.

The authors provide a correlation matrix on surface and expression, but should that not be more than 50% in the best case? Clusters 2,7 and 8 that are bona fide macrophage clusters do actually not very well positively correlate with A-G, at least in the opinion of this reviewer. It looks more like these are the remaining ones as 1 and 11 are more linked to F,E,G? The authors might add additional information to provide less complex information to the reader to allow for understanding of the overlap. As proteins will become central for staining of tissues, that information is critical.

Figure 2 is more like a supplementary figure rather than novel information in itself. It is an interesting analysis, but any data to test or further confirm this information is not provided. And, as the authors imply that the terms and interactions listed are functionally relevant rather than simply grouping determinants, again either show additional functional data to support the analysis or list them as very interesting supplementary material that await further experimental confirmation with respect to function.

Figure 2D: In the analysis of potential ligand-receptor interactions, they use the term LT-HSC (line 254). However, the population for the receptors in Figure 2D is shown as CD150+CD48+, which doesn't represent the immunotypic CD150+CD48-LSK LT-HSC population described in the methods. What is the cell-type shown in these analyses?

Figure 3: In general, as only the CD150 antibody is used to identify HSCs, the term HSCs for CD150+ cells is not correct. Progenitor and megakaryocytic cell populations are also CD150+. The high number of CD150+ cells in the diagram (Figure 3C) likely reflects this heterogeneity of the CD150+ cells, as there should be fewer HSCs. The term HSC/MPPs might be more correct here, which though then implies that data is not fully HSC centered. The authors might want to resolve that. Please also provide readable scale bars for the histology images. In Figure 3B, please also show the SMA (vessel) staining in the enlargements.

The images in Figure 3E are difficult to follow, but are central to the publication, as they investigate the special relationship between macrophages from the clusters and HSCs. CD150 is in black? What is white? It looks like that some of the HSCs are F4/80 positive? HSCs are much larger than macrophages and multinucleated? So it remains not clear whether what is identified as HSCs is indeed HSCs or what is what. Macrophages identified in Figure 3 look different from the ones identified in S3C? The authors might want to really clarify these questions and also support their claims with additional new data.

And, would it be possible to stain for cluster A-D in one single staining, to see the extent of overlap/exclusivity?

The image for cluster C is likely not representative as the frequency of HSCs is similar to cluster A,B and D, but according to the claims there should be more HSCs. But see also comment on Figure 3F below.

If Figure 3F is interpreted correctly, most of the HSCs analyzed do not have a cell from the cluster close by. That would mean that most HSCs in the fetal liver are not surrounded by macrophages from cluster A-D. That would also mean that likely the niche function of macrophages for HSCs in the fetal liver is rather limited, which would align with the finding that specifically depleting macrophages in the fetal liver (Figure 5).

Figure legend 3A: please add Iba1. Also, change the order of the row descriptions or the images.

Figure 4: The depletion of macrophages (Figure 4B), which are highly abundant in FL tissue (Figure 3B,C) do not lead to changes in tissue architecture (HE experiments, Figure 4D) or why is there a reduction in overall cell number, but not in CD45+ cells? That is somewhat surprising, as this reviewer might actually expect a consequence upon deletion that is larger than a reduction of enucleated erythrocytes from 70 to 50%. How do the authors explain that? What is the longer-term (development) consequence of Tnfrsf11a driven deletion of Spi1 positive cells? Is there any?

Figure 5: To the best of the knowledge of this reviewer, while HSCs in E14.5 FL follow the SLAM code, whether all the types of progenitors identified by gating like adult progenitors indeed show in FL the potential that is associated with that gating strategy in the adults has not been unequivocally demonstrated. The authors might at least comment on that in the manuscript.

There seems to be an upregulation of genes linked to myeloid cell differentiation in KO cells. Were any genes associated with lymphoid differentiation also affected?

Figure 6: Why did the author not use simple co-culture experiments to test an influence of cluster A-D macrophages on HSCs? That would deliver direct interaction data.

Figure 6B. It is not clear to this reviewer how the cells have been gated, and how neutrophils have been identified etc.

Figure 6C: Technically, there is no effect on the GM population in these mice. While there is a clear trend, the authors really need to come up with additional data to test this likely difference, as this data made it into the title, and is the only functional difference they claim that lack of macrophages in fetal liver has on HSCs. An event that might be indirect.

---

## [Author Response]

Essential revisions (for the authors):The reviewers concluded that although the strength of the study lies in the characterization and categorization of distinct FL macrophages, and in the fate-mapping aspect, the conclusion that macrophages are regulating the hematopoietic stem cell niche and granulopoiesis is not well-supported by the data as presented. Either transplantation or co-culture experiments at a minimum would be necessary to demonstrate an effect on hematopoiesis and lineage differentiation. Alternatively, strongly adapting the language/the conclusion on the influence of the macrophages on HSCs in the fetal liver might improve the study.

We would like to thank all reviewers for the thorough evaluation of the manuscript and have now conducted a series of new experiments that strengthen the conclusion that macrophages are regulating the hematopoietic stem cell niche and granulopoiesis.

Reviewer #1 (Recommendations for the authors):Based on initial clustering, it seems that many other macrophages within the fetal liver are excluded from further downstream analysis. It would have been very interesting and important to investigate both fate-mapping and gene expression signatures in other clusters besides those that specifically align with pMAC signatures.

The scRNA-seq analysis was performed to find distinct macrophage populations and surface markers that we could use to prove on the protein level that these populations are not transient cellular states. Therefore, we performed a quite stringent analysis of the scRNA-seq data to exclude any other myeloid progenitors (Figure 1 —figure supplement 1). The robustness of the reduction from 11 to 4 clusters, based on the pMac signature, is underpinned by the thorough validation from a previous comprehensive study (Mass et al. Science 2016). This signature reliably identifies macrophages, ensuring the robustness of our initial selection step, which is crucial for the integrity of all downstream analyses. Moreover, the cell types identities were investigated using the marker genes per each cluster to assure the selection of the cells. The cells not included in the further analysis were identified as other cell types such as DCs (expression of *Cd74, Cts3*), pDCs (expression of *Siglech, Irf8*), granulocytes (expression of *Mpo, Elane*). We have now added the cell type identification on the UMAP in Figure 1 —figure supplement 1B.

Further, in the subsequent reduction we excluded what can be likely identified as myeloid progenitor cells and/or a contamination by monocytes/granulocytes in previous clusters as evidenced by their gene expression profiles of *Ly6c2, Ly6g, Cxcr2*, and *Cd33* but lack of bona-fide macrophage markers. These exclusions align with our objective of investigating macrophage populations and their roles within the fetal liver's hematopoietic environment.

Fate-mapping of potential different macrophage precursors is an interesting point. However, this was not the aim of this study and, therefore, would go beyond its scope.

The authors might have detected the effects of macrophage depletion on HSC function if they had isolated LT-HSCs and performed transplantation assays. This would have provided much stronger evidence supporting the author's conclusion of granulocytic bias. Colony-forming assays, as presented, are not convincing of the role of fetal macrophages in regulating HSC output or function, particularly in the absence of any effects in vivo in the conditional deletion model.

We agree with the reviewer and thus repeated and performed additional experiments (see response above).

Also, it may have been too early for LT-HSCs to produce granulocytes, and for them to significantly accumulate in the FL. This could be directly tested using the fate-mapping models on hand.

This is an interesting point, which we have analysed in our initial dataset but did not include in the manuscript since this adds an additional layer of hematopoiesis that is beyond the scope of the current manuscript. Our fate-mapping of neutrophils shows that all of them are labelled by the Cxcr4-CreERT2 model but only half by the Ms4a3-Cre model. This suggests that the production of granulocytes/neutrophils is happening in at least 3 waves: 1. Yolk sac EMP-derived (labelled by Csf1r-MeriCreMer, see Gomez-Perdiguero et al., 2015, Nature), 2. Definitive HSC-derived (labelled by Ms4a3-Cre and Cxcr4-CreERT2) and 3. “Transient” HSC-derived (labelled only by Cxcr4-CreERT2). The latter hypothesis is supported by a recent paper stating that pre-HSPCs give rise to many progenitors in the E14.5 liver (Yokomizo et al. Nature 2022). However, due to the Ms4a3-Cre model we concluded that at least half of the neutrophils are LT-HSC derived. Furthermore, our new dataset showing the significant increase of neutrophils still being present at E18.5 strengthen the hypothesis that macrophages contribute to the granulopoiesis potential of LT-HSCs which are the major source of neutrophils at this time point.

In general, some of the data could be more clearly presented. For example, frequencies should be shown in 1i. Data in Figure 3E don't represent data in Figure 3F – e.g., there appear to be as many macrophages of each subtype in proximity to each HSC in 3E. In 3C, Cluster C is not visible. In Figure 6, differences in cluster size between KO and WT should be quantified. These minor changes would help clarify some of the data.

We have addressed these points in our response above.

Reviewer #2 (Recommendations for the authors):Figure 1 and Figure S1. The analysis and identification of macrophages in the fetal liver is strong. The initial selection step reduces the number of clusters further analyzed from 11 to 4, simply based on the pMac signature, plus a subsequent reduction (Figure S1D). How robust is that reduction, as strongly influences all downstream analyses and reduces the number of cells for subsequent analyses? What might then be cells not included?Is the analysis robust enough to then include all types of macrophages, or the majority populations?

We appreciate the reviewer's comment and have improved the representation of our data accordingly. The robustness of the reduction from 11 to 4 clusters, based on the pMac signature, is underpinned by the thorough validation from a previous comprehensive study (Mass et al. Science 2016). This signature reliably identifies macrophages, ensuring the robustness of our initial selection step, which is crucial for the integrity of all downstream analyses. Moreover, the cell types identities were investigated using the marker genes per each cluster to assure the selection of the cells. The cells not included in the further analysis were identified as other cell types such as DCs (expression of *Cd74, Cts3*), pDCs (expression of *Siglech, Irf8*), granulocytes (expression of *Mpo, Elane*). We have now added the cell type identification on the UMAP in Figure 1 —figure supplement 1B.

Further, in the subsequent reduction we excluded what can be likely identified as myeloid progenitor cells and/or a contamination by monocytes/granulocytes in previous clusters as evidenced by their gene expression profiles of *Ly6c2, Ly6g, Cxcr2*, and *Cd33* but lack of bona-fide macrophage markers. These exclusions align with our objective of investigating macrophage populations and their roles within the fetal liver's hematopoietic environment.

To this reviewer, the presentation of the alignment of the flow-based analysis (informed by the gene expression data) that was also reduced in complexity aligned that likely well to the gene expression clusters could be further improved and supported by additional information and steps much better put into perspective. This alignment is critical as protein and gene-based clustering seem to align, which is central due to the high level of reduction that was applied in obtaining the results.

The analysis appears robust in identifying and including the majority populations of macrophages within the fetal liver. Due to the stringent selection criteria based on the pMac signature, we have captured all the major types of macrophages or transitional states that could be present in our analysis. Thus, while the analysis captures the dominant macrophage populations and provides a detailed insight into their roles, it still should be able to pick macrophage subpopulations or states that might be transitional in the analyzed clusters. Based on our methodological approach, we believe we have captured all possible macrophage populations within the fetal liver.

The authors provide a correlation matrix on surface and expression, but should that not be more than 50% in the best case? Clusters 2,7 and 8 that are bona fide macrophage clusters do actually not very well positively correlate with A-G, at least in the opinion of this reviewer. It looks more like these are the remaining ones as 1 and 11 are more linked to F,E,G? The authors might add additional information to provide less complex information to the reader to allow for understanding of the overlap. As proteins will become central for staining of tissues, that information is critical.

The correlation percentage depicted is indeed modest; however, it's notable given the few features utilized and the inherent complexity and disparate nature of gene and protein expression data. Achieving over 50% correlation is challenging under these circumstances. Our analysis framework was meticulously designed to accurately reflect the relationships between gene expression clusters and protein marker clusters A-G, despite these challenges. The achieved percentage, although not exceedingly high, is significant given the disparate nature of the data and the few features utilized in the analysis. This correlation, along with the robustness of our methodological approach, underscores the reliability of our findings in representing the macrophage populations within the fetal liver and their roles in the hematopoietic environment. To start with less complex information for the reader, the correlation matrix was newly scaled and moved to the Figure 1 —figure supplement 2.

Figure 2 is more like a supplementary figure rather than novel information in itself. It is an interesting analysis, but any data to test or further confirm this information is not provided. And, as the authors imply that the terms and interactions listed are functionally relevant rather than simply grouping determinants, again either show additional functional data to support the analysis or list them as very interesting supplementary material that await further experimental confirmation with respect to function.

We agree with the reviewer that this Figure is characterizing the signatures of macrophage subpopulations in more detail without functional data. However, we believe that proving the functional importance of these paracrine factors, beyond the listed factors that have already been shown to be essential for LT-HSC function (see discussion), is beyond the scope of this manuscript. We aim at addressing this point in future studies. Nevertheless, we would not want this data to “disappear” into the supplements as macrophage-derived paracrine signals shown in the network are partially novel and may be interesting for the community.

Figure 2D: In the analysis of potential ligand-receptor interactions, they use the term LT-HSC (line 254). However, the population for the receptors in Figure 2D is shown as CD150+CD48+, which doesn't represent the immunotypic CD150+CD48-LSK LT-HSC population described in the methods. What is the cell-type shown in these analyses?

We are sorry for the mistake. In this figure, we have sorted (Figure 6 —figure supplement 1A) and analysed CD150+CD48-LSK LT-HSC

Figure 3: In general, as only the CD150 antibody is used to identify HSCs, the term HSCs for CD150+ cells is not correct. Progenitor and megakaryocytic cell populations are also CD150+. The high number of CD150+ cells in the diagram (Figure 3C) likely reflects this heterogeneity of the CD150+ cells, as there should be fewer HSCs. The term HSC/MPPs might be more correct here, which though then implies that data is not fully HSC centered. The authors might want to resolve that.

We thank the reviewer for this comment. We have now added CD41 into the CODEX panel and performed a new staining to distinguish between LT-HSCs and megakaryocytes/megakaryocyte progenitors using the Cxcr4-CreERT; Rosa26-YFP fate-mapping mice, which target definitive hematopoiesis, and thus also LT-HSCs. However, our quantification showed almost no CD150+ CD41+ double-positive cells (Figure 4 —figure supplement 2D, E). Together with the gating strategy shown to sort LT-HSCs form E15 livers in the study by Ghosn et al. 2016 (now cited in the respective section about CD41) we therefore concluded that cells identified by CD150 expression are mostly LT-HSCs. We have also included a paragraph on the fact that fetal HSC/MPP may be different in terms of their surface expression markers and the differentiation potential seen in adult bone marrow (lines 527-529).

Please also provide readable scale bars for the histology images. In Figure 3B, please also show the SMA (vessel) staining in the enlargements.

We provide scale bars for all histology images, lengths are indicated in the figures and/or the legends. SMA was added in Figure 3B (now Figure 4B)

The images in Figure 3E are difficult to follow, but are central to the publication, as they investigate the special relationship between macrophages from the clusters and HSCs. CD150 is in black? What is white? It looks like that some of the HSCs are F4/80 positive? HSCs are much larger than macrophages and multinucleated? So it remains not clear whether what is identified as HSCs is indeed HSCs or what is what. Macrophages identified in Figure 3 look different from the ones identified in S3C? The authors might want to really clarify these questions and also support their claims with additional new data.

To improve the data of macrophage-HSC interaction we have performed 3D reconstructions and quantified the distance of CD150+ and Iba1+ cells in 3D (new Figure 3C-E) as the thin cryosectioning used for CODEX is not suitable to reconstruct these interactions properly (see also lines 328-331). Thus, Figure 3E was not able and also not meant to represent data shown in Figure 3F (now Figure 4E and 4F). Figure 3E is just meant to show examples of all clusters sitting in proximity to CD150+ HSCs. Black font (CD150) is indicating the white signal, we added this information in the legend. Since the CODEX images are too large to image them on a confocal microscope, co-localization of signals indicates a clear neighbourhood of cells.

We are not sure what the reviewer means with “Macrophages identified in Figure 3 look different from the ones identified in S3C”. In Figure S3C (now Figure 4 —figure supplement 1) we moved to a region of the liver with many Ter119+ cells, which are densely packed and where Ter119+ cells surrounding macrophages. This is likely why the appearance changes from the representations shown in Figure 3 (now Figure 4).

And, would it be possible to stain for cluster A-D in one single staining, to see the extent of overlap/exclusivity?

We now show co-stainings for each combination of the clusters (CD169/CD106/CD206/Ter119) in Figure 4 —figure supplement 2C with most of the clusters showing only little overlap (Pearson correlation <0.3).

The image for cluster C is likely not representative as the frequency of HSCs is similar to cluster A,B and D, but according to the claims there should be more HSCs. But see also comment on Figure 3F below.If Figure 3F is interpreted correctly, most of the HSCs analyzed do not have a cell from the cluster close by. That would mean that most HSCs in the fetal liver are not surrounded by macrophages from cluster A-D. That would also mean that likely the niche function of macrophages for HSCs in the fetal liver is rather limited, which would align with the finding that specifically depleting macrophages in the fetal liver (Figure 5).

We apologize if we did not represent the data in a way that allowed clear interpretation. We tried to address this point in the previous manuscript: “Of note, the tissue analysed via CODEX represents only cellular neighbourhoods in X and Y due to the thin sectioning technique (5 μm) and, thus, does not take neighbouring cells in the Z plane into account. This leads to an underestimation of macrophage-HSC interactions, as indicated by F4/80+ filopodia extending towards CD150+ cells in almost all cases (Figure 3E).” Furthermore, there are many more macrophages in the tissue than HSCs so that we would not expect a certain population to be interacting with more than one CD150+ cell.

We have now addressed this limitation that is inherent to CODEX technology by analysing whole mount stainings of fetal livers. We show that indeed most CD150+ HSCs are within a ~1.4-1.8µm radius of Iba1+ macrophages and that this distance is significantly shorter than for any other random cell (new Figure 3C-E).

Figure legend 3A: please add Iba1. Also, change the order of the row descriptions or the images.

We have added Iba1 and changed the order as suggested by the reviewer.

Figure 4: The depletion of macrophages (Figure 4B), which are highly abundant in FL tissue (Figure 3B,C) do not lead to changes in tissue architecture (HE experiments, Figure 4D) or why is there a reduction in overall cell number, but not in CD45+ cells? That is somewhat surprising, as this reviewer might actually expect a consequence upon deletion that is larger than a reduction of enucleated erythrocytes from 70 to 50%. How do the authors explain that?

To confirm that there are no big cellular changes beyond the lack of macrophages, as suggested by the HE stainings, we analysed more fetal livers from both genotypes. We do not expect that the overall structure of the liver would change due to the lack of approx. 250.0000 macrophages (See Figure 5A and 7C) that measure via flow cytometry. The changes of enuclated erythroblasts have been analysed in the blood, while the data in Figure 4B-E (now Figure 5B-E) are showing fetal livers. We have clarified that now in the text.

What is the longer-term (development) consequence of Tnfrsf11a driven deletion of Spi1 positive cells? Is there any?

We have now also analysed E16.5 and E18.5 fetal livers from Tnfrsf11a-Cre X Spi1fl/fl embryos. We see that the empty macrophage niche is being repopulated and that the increased production of granulocytes remains stable throughout embryogenesis.

From our current and previous studies (Jacome-Galarza et al. 2019, https://doi.org/10.1038/s41586-019-1105-7; Cox et al. 2021, Science https://doi.org/10.1126/science.abe9383) we know that Tnfrsf11a-Cre X Spi1fl/fl animals are born osteopetrotic and survive only until max. 3-4 weeks of age.

Figure 5: To the best of the knowledge of this reviewer, while HSCs in E14.5 FL follow the SLAM code, whether all the types of progenitors identified by gating like adult progenitors indeed show in FL the potential that is associated with that gating strategy in the adults has not been unequivocally demonstrated. The authors might at least comment on that in the manuscript.

This is a fair point, and we agree that there has not been a systematic analysis addressing the potential of HCS/MPPs in the fetal liver. We added this point in the discussion (lines 527-532).

There seems to be an upregulation of genes linked to myeloid cell differentiation in KO cells. Were any genes associated with lymphoid differentiation also affected?

We indeed identified a set of genes associated with lymphocyte differentiation that were upregulated in our study. Among them some have been already studied for their role in lymphocyte proliferation and differentiation. For example, Bcl2 (https://www.nature.com/articles/s41598-019-41247-5), Braf https://www.annalsofoncology.org/article/S0923-7534(19)34279-6/fulltext ,

Gata3 (https://www.frontiersin.org/articles/10.3389/fimmu.2022.975958/full , https://ashpublications.org/blood/article/121/9/1534/138823/E2A-transcription-factors-limit-expression-of), Clcf1 (https://journals.aai.org/jimmunol/article/208/1_Supplement/46.13/235942/Characterization-of-a-new-cytokine-complex-from)

Furthermore, we also identified other genes contributing to the lymphoid differentiation based on our GO analysis (Supplementary File 4). These genes include Abl1, Cdh17, Clec4e, Dock2, Egr1, Foxp1, Hdac9, Il1rl2, Inpp5d, Irf1, Lilrb4b, Lmbr1l, Nfkbid, Pbx1, Pglyrp2, Plcg2, Pou2f2, Ptprc, Rc3h2, Runx2, Stat3, Tox, Zfp608, and Zfp609.

Some of the listed genes are rather generic and regulate other processes in addition to lymphoid differentiation. Due to the low output of lymphoid cells at E14.5 and the unaltered CLP numbers in the fetal liver, we did not investigate this point further.

Figure 6: Why did the author not use simple co-culture experiments to test an influence of cluster A-D macrophages on HSCs? That would deliver direct interaction data.

Macrophages rapidly de-differentiate in cell culture conditions (DOI:https://doi.org/10.1016/j.neuron.2017.04.043). Therefore, with the need of lengthy sorting to isolate macrophage subpopulations, we decided initially not to perform co-culture experiments since the impact of the subpopulations could not be addressed by this experiment in our opinion. Instead, we aimed at defining macrophages as an essential LT-HSC niche cell in the first place. Future experiments knocking out specific paracrine factors stemming from distinct macrophages populations will inform on the impact of these populations on LT-HSC functionality.

Figure 6B. It is not clear to this reviewer how the cells have been gated, and how neutrophils have been identified etc.

The gating for neutrophils and all other cells is shown in Figure 7 —figure supplement 1.

Figure 6C: Technically, there is no effect on the GM population in these mice. While there is a clear trend, the authors really need to come up with additional data to test this likely difference, as this data made it into the title, and is the only functional difference they claim that lack of macrophages in fetal liver has on HSCs. An event that might be indirect.

We agree with the reviewer and therefore performed new experiments to show that increase of granulopoiesis is a stable result when HSCs are isolated form livers lacking macrophages:

We have performed the colony-forming assay again with n=5 embryos per genotype that were harvested on the same day, which resulted in a similar phenotype as before, with the differences of GM colonies now being significant.To strengthen the point that the transient lack of macrophages when HSCs arrive in the fetal liver leads to their reprograming, we included flow cytometry data from E16.5 and E18.5 where we still see an increase of neutrophils in the fetal liver, despite the fact that macrophages are repopulating the empty niche (Figure 7E, F).To show that this is a cell-intrinsic effect, we have performed adoptive transfer experiments supporting our claim that loss of macrophages reprograms HSCs toward the granulocytic lineage (Figure 7H, I)